**RESEARCH**

# Deep sequencing reveals a DAP1 regulatory haplotype that potentiates autoimmunity in systemic lupus erythematosus

Prithvi Raj[1*†] (iD), Ran Song[1†], Honglin Zhu[2], Linley Riediger[1], Dong-Jae Jun[3], Chaoying Liang[1], Carlos Arana[1], Bo Zhang[1], Yajing Gao[1], Benjamin E. Wakeland[1], Igor Dozmorov[1], Jinchun Zhou[1], Jennifer A. Kelly[4], Bernard R. Lauwerys[5], Joel M. Guthridge[4], Nancy J. Olsen[6], Swapan K. Nath[4], Chandrashekhar Pasare[1], Nicolai van Oers[1], Gary Gilkeson[7], Betty P. Tsao[7], Patrick M. Gaffney[4], Peter K. Gregersen[8], Judith A. James[4], Xiaoxia Zuo[2*], David R. Karp[9], Quan-Zhen Li[1*] and Edward K. Wakeland[1*]

* Correspondence: prithvi.raj@utsouthwestern.edu; susanzuo@csu.edu.cn; quan.li@utsouthwestern.edu; edward.wakeland@utsouthwestern.edu
†Prithvi Raj and Ran Song contributed equally to this work.
[1]Department of Immunology, University of Texas Southwestern Medical Center, Dallas, TX 75390, USA
[2]Department of Rheumatology and Immunology, Xiangya Hospital, Central South University, Changsha 410008, China
Full list of author information is available at the end of the article

## Abstract

**Background:** Systemic lupus erythematosus (SLE) is a clinically heterogeneous autoimmune disease characterized by the development of anti-nuclear antibodies. Susceptibility to SLE is multifactorial, with a combination of genetic and environmental risk factors contributing to disease development. Like other polygenic diseases, a significant proportion of estimated SLE heritability is not accounted for by common disease alleles analyzed by SNP array-based GWASs. Death-associated protein 1 (DAP1) was implicated as a candidate gene in a previous familial linkage study of SLE and rheumatoid arthritis, but the association has not been explored further.

**Results:** We perform deep sequencing across the DAP1 genomic segment in 2032 SLE patients, and healthy controls, and discover a low-frequency functional haplotype strongly associated with SLE risk in multiple ethnicities. We find multiple cis-eQTLs embedded in a risk haplotype that progressively downregulates DAP1 transcription in immune cells. Decreased DAP1 transcription results in reduced DAP1 protein in peripheral blood mononuclear cells, monocytes, and lymphoblastoid cell lines, leading to enhanced autophagic flux in immune cells expressing the DAP1 risk haplotype. Patients with DAP1 risk allele exhibit significantly higher autoantibody titers and altered expression of the immune system, autophagy, and apoptosis pathway transcripts, indicating that the DAP1 risk allele mediates enhanced autophagy, leading to the survival of autoreactive lymphocytes and increased autoantibody.

**Conclusions:** We demonstrate how targeted sequencing captures low-frequency functional risk alleles that are missed by SNP array-based studies. SLE patients with the DAP1 genotype have distinct autoantibody and transcription profiles, supporting the dissection of SLE heterogeneity by genetic analysis.

**Keywords:** SLE, Sequencing, DAP1, SNPs, Haplotype, RNA-Seq, Autophagy

## Introduction

Systemic lupus erythematosus (OMIM 152,700) is a clinically heterogeneous auto-immune disease characterized by the development of anti-nuclear antibodies (ANA) and the deposition of immune complexes throughout the body [1–5]. Disease manifestations are highly complex and variable within patient populations, with disease classification normally based on the presence of multiple clinical and laboratory phenotypes [6, 7]. The complex etiology and pathogenesis of lupus complicates its diagnosis and is one of the greatest challenges to the development of effective therapies.

Susceptibility to SLE is multifactorial, with a combination of genetic and environmental risk factors contributing to disease development [5, 8, 9]. Many genome-wide association studies (GWAS) of SLE have been published over the past decades, and a multitude of susceptibility loci have been identified [10–16]. In addition, three recent meta-analyses of SLE GWAS datasets containing European American, African Americans, Hispanics, and several Asian cohorts identified over 90 risk loci [17–19]. In each of these studies, roughly 40–50% of the identified loci reached genome-wide significance ($p < 5 \times 10e{-}8$), with roughly 30% of the rest reaching suggestive levels ($p < 5 \times 10e{-}5$) and the remainder representing candidate genes ($p < 10e{-}3$). Thus, about 45 SLE risk loci with genome-wide significance are prevalent in the global SLE patient population and the remainder are detected with suggestive or candidate gene association levels, often in a single or a limited number of patient cohorts. As with all polygenic diseases, a significant proportion of the estimated disease heritability is not accounted for by these "common" disease alleles [20–24].

Our recent work has focused on identifying the causal variants that underlie the functional properties of SLE risk alleles with the goal of better understanding the dysregulated molecular pathways that lead to disease. We have implemented a population sequencing strategy for the identification of the causal functional variants in SLE risk loci. In our first population sequencing study, we identified 124,552 variants within the LD blocks of 28 SLE risk loci in 1775 SLE patients and controls [25]. Further analysis identified 1204 common (MAF > 0.05) variants within this sequence dataset that had similar or increased levels of disease association in comparison to previously published GWAS tagging SNPs for the same risk locus [25]. Most of these variants modified regulatory segments that controlled gene transcription and/or chromatin structure, with only a few (7 of the 1204) directly impacting the protein structures of relevant candidate genes. These disease-associated regulatory alleles modulated the transcription of key immune system genes (i.e., *HLA-DR*, *HLA-DQ*, *STAT1/4*, *IRF5*, etc.), indicating that predisposition to autoimmunity often reflects transcriptional variations in immunoregulatory molecules that impact the activation thresholds of immune signaling pathways.

A variety of molecular pathways in the innate and adaptive immune system are dysregulated in SLE patients, and many of these are impacted by SLE risk alleles [26, 27]. For example, genetic variations in multiple members of the autophagy pathway have been associated with risk for SLE. Variations in *ATG5*, a key component of the autophagy pathway, were associated with SLE susceptibility from the beginning of GWAS studies of European American [10], and several recent studies in Asian cohorts have identified four additional autophagy pathway genes (*ATG16L2*, *DRAM1*, *CDKN1B*, and *CLC16A*) with genome-wide significant associations and seven other autophagy genes with candidate levels of association with SLE [10, 28–32]. The autophagy pathway has been shown to impact SLE through the LC3-associated phagocytosis (LAP) pathway,

which is a key cellular process in antigen presentation and in promoting the longevity of immune cells by decreasing their susceptibility to apoptosis [33–38].

Death-associated protein 1 (DAP1) is a 15 kd proline-rich protein that has been shown to function as a negative regulator of autophagy [39, 40]. DAP1 is a substrate of mTORC1, which is a serine/threonine kinase that "senses" the metabolic status of a cell and limits the initiation of autophagy in nutrient-rich conditions. The majority of DAP1 protein is maintained in an inactive state by mTORC1 phosphorylation in nutrient-rich conditions; however, DAP1 becomes dephosphorylated and active when mTORC1 kinase activity is attenuated due to nutrient depletion [41, 42]. Decreased mTORC*1* kinase activity initiates autophagy (through ULK1) and simultaneously causes DAP1 to become an active autophagy suppressor, consistent with a potential role of DAP1 as a "brake" that will diminish signaling through this pathway and avoid excessive autophagy. Thus, dysregulation of DAP1 expression could significantly modulate the intensity of autophagy produced in cells under activating conditions. Recent GWAS meta-analyses of inflammatory bowel disease (IBD) associated a variant in DAP1 with IBD at genome-wide significance [43, 44]. Given our extension data set and large population cohort, we decided to directly address the association of this locus with autoimmunity in the present study. It is true that previous investigations, based on genotyping arrays such as Immunochipv.1, observed a relatively weak association between DAP1 and SLE. But we hypothesized that the poor association was due to the absence of key functional variants on these arrays. Typically, SNP arrays only capture a few of the common frequency variants or tag SNPs within a given locus as was the case with the DAP1 gene, which was represented by only 12 markers on Immunochipv.1. Much stronger data concerning the functional variants of DAP1 that drive the association statistics was needed. Although *DAP1* has not yet been associated with SLE by *GWAS*, it was implicated as a candidate gene in a familial linkage study of SLE and RA [45, 46] and reached marginal candidate gene status in our recent Immunochipv1 study of SLE [18]. But since these studies were based on fewer number of tag SNPs, they did not allow assessment of the contribution of low frequency or rare alleles in susceptibility to autoimmune or rheumatic diseases. Therefore, we applied targeted and deep sequencing approach to capture all common and rare genetic variants and then assess their potential association with SLE. Our hypothesis is that low-frequency functional variants that are missed on SNP arrays underlie previously observed association of *DAP1* with SLE.

So, here we describe a genomic analysis of the *DAP1* locus in 1221 SLE patients and 811 healthy controls from different ethnic backgrounds. We discover extensive genetic diversity in this locus and demonstrate that a haplotype containing a specific constellation of regulatory variants is a potent, low-frequency, risk allele for SLE in all three ethnic groups tested. This risk haplotype downregulates the transcription of *DAP1* in multiple immune cell lineages, leading to increased autophagy, diminished apoptosis, increased humoral autoimmunity, and increased risk for discoid rash. Our analyses provide insights into how specific combinations of regulatory polymorphisms impact disease phenotypes and demonstrate the potential of genetic analysis to stratify SLE patients into subgroups with distinct disease characteristics.

## Results

### Genomic diversity of the *DAP1* segment and SLE associations

We initially assessed the association of *DAP1* polymorphisms with SLE using data produced as part of our recent genomic scan of SLE with the Immunochipv.1 array [18]. Only 12 SNPs from the *DAP1* region were included on the Immunochipv.1. As shown in Additional file 1: Table S1, very weak associations were found between *DAP1* and SLE in an analysis of 1700 SLE patients and 2108 healthy European Americans. Only three of the twelve SNPs assayed showed any association ($p < 0.05$, FDR) and the peak association detected was with rs2918391 (OR 1.1, $p < 3.3 \times 10e{-}3$).

A 102-kb genomic segment (Chr5:10678000-10780000 hg19) containing the *DAP1* locus was included in the SLEv.1 targeted sequencing panel that we previously used to sequence and analyze 28 SLE risk loci [25]. As shown in Fig. 1a, we produced high-quality DNA sequence from this region with an average depth of 153.2-fold coverage for 773 SLE patients and 576 ethnicity-matched healthy European Americans. Sequence

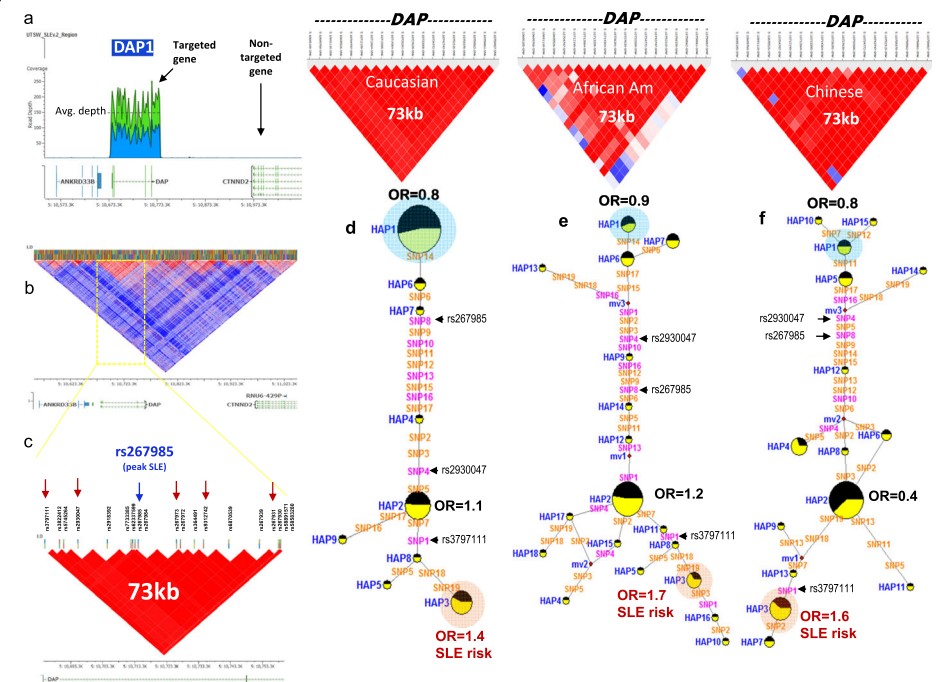

**Fig. 1** SLE-associated *DAP1* haplotype in European American, African American, and Asian populations. **a** The targeted sequencing strategy and read depth across *DAP1* gene locus on chromosome 5. **b**, **c** The overall LD (linkage disequilibrium) structure across *DAP1* region; LD blocks shown in panels **b** and **c** are generated based on Caucasian data from the 1000 Genome Project and Caucasian SLE and control samples from the present study, respectively. **b** Show 73-kb block that contains top 19 SLE-associated *DAP1* variants. Location of the peak SLE-associated SNP rs267985 is shown in **c**. **d** Median-joining (MJ) network analysis on most common *DAP1* haplotypes in European American SLEs and healthy controls. Spheres (termed nodes) represent the locations of each haplotype (Additional file 1: Table S6) within the network, and the size of the node is proportional to the overall frequency of that haplotype in the dataset. Each node is overlaid with a pie chart that reflects the frequency of that haplotype in the SLE group (yellow) versus healthy group (black). The lines connecting the nodes are labeled with the variants that distinguish the connected nodes, and the length is proportional to the number of variants. Odds ratio (OR) is shown for the two most significant alleles. Red highlighted nodes indicate SLE risk clades. **e** Median-joining (MJ) network analysis on most common *DAP1* haplotypes in African American SLEs and healthy controls. **f** Median-joining (MJ) network analysis on most common *DAP1* haplotypes in Asian SLEs and healthy controls

variance analysis identified 2094 high-quality sequence variants throughout this segment of which 1764 were low frequency (MAF < 0.05) and 330 were common (Additional file 1: Table S2a and S2b). These results illustrate that the *DAP1* genomic region is extremely polymorphic, having an average of 20.5 variants/kb of which 3.23 were common within the European American population. As shown in Fig. 1b, c, linkage disequilibrium analysis identified a 73-kb LD block encompassing the entire *DAP1* gene in European Americans. A genetic association analysis with the 330 common variants identified 39 with statistically significant ($p < 0.05$, FDR) association with SLE in European Americans. The strongest association in European Americans was observed with SNP rs267985 [OR (95% CI) = 1.38 (1.1–1.6), $p = 1.81e–04$], which is an intronic variant within *DAP1* with strong regulatory potential based on the ENCODE and RegulomeDB databases [47, 48].

Additional file 2: Fig.S1a plots the association strength of the common SLE variants and delineates between variants that are potentially functional (yellow points, ENCODE) or currently unannotated (blue). These data illustrate that the majority of the SLE-associated variants are potentially functional and that all SLE-associated variants are within the 73-kb LD block, which also includes the IBD-associated variant described previously [43, 44]. Conditional analysis on peak SNP rs267985 removes all the disease association for SLE (Additional file 2: Fig. S1b-c), indicating that only one signal of association with SLE occurs within this genomic segment. These results demonstrate the presence of a wealth of variations within the *DAP1* LD block and provide evidence of a much stronger association with SLE than could be detected via analysis with the Immunochipv.1.

The properties of nineteen SNPs that were most strongly associated with SLE in this initial analysis (all with $p \leq 3.45e–03$ in European Americans) are summarized in Additional file 1: Table S3 and Additional file 2: Fig.S2. As shown, none of these variants impacted DAP1 protein structure, but all were in non-coding regions that are associated with the regulation of transcription and/or chromatin structure. To test the observed association of these *DAP1* variants with SLE in other ethnic populations, we used our SLEv2 targeting array to sequence *DAP1* in 363 African Americans SLEs and control individuals and 320 Chinese SLEs and control individuals. This sequence analysis revealed a total of 1117 new variants, of which 254 were common and 863 were low frequency or rare (Additional file 1: Table S4). These results further document the extensive diversity of *DAP1* in the global population and illustrate the expanded variant dataset that is obtained by genome targeted sequencing of risk loci. As shown in Additional file 1: Table S5, Chochran-Mantel-Haenszel (CMH) analysis of these 19 SNPs in all three ethnic populations detected significantly stronger SLE associations. The peak association in the CMH analysis was with SNP7 ($p < 8.26e–06$), while with SNP8, the peak association in European Americans was also more strongly associated ($p < 3.8e–05$). In addition, SNP1, SNP18, and SNP19 all had strong associations with SLE ($p < 5e–05$) in the CMH analysis. We further performed trans-ethnic meta-analysis on top 19 SLE-associated variants using MR-MEGA Program [49, 50]. The results further support our original findings. Consistent with CMH analysis, SNP1 (rs3797111) and SNP19 (rs58583280) show strong association with SLE. Also, SNP2 (rs3822412), SNP7 (rs62337599), SNP8 (rs267985), and SNP12 (rs364491) show strong association. These results are consistent with the fact that SNP1, SNP7, and SNP18 define the strongest

risk haplotype 3 (HAP3). We also performed genetic association analysis on low-frequency variants using the KBAC (kernel-based adaptive cluster) method. This method uses counts of multi-marker genotypes to perform a special multivariate case/control test to determine their association with the disease phenotype. Results showed significant (*t* test, $p = 0.035$) association for DAP genotypes with SLE.

All of the 19 SLE-associated variants were in strong LD (D' > 0.8) (Fig. 1b, c), and haplotype analysis using Haploview detected three common haplotypes that accounted for over 80% of the chromosomes in the combined study population (Additional file 1: Table S6). Figure 1 panels d, e, and f present the sequence divergence networks that are defined in each study population utilizing the median neighbor joining (MJ) algorithm [51]. For MJ diagrams, the spheres (termed nodes) represent individual haplotypes in the network and their size is proportional to their frequency in the study population. The pie charts overlaid on each node represent the relative frequency of that haplotype in SLE cases (yellow) and healthy controls (black). The individual SNPs that distinguish each node are listed along the line that connects them. These networks are progressive, such that the two nodes at opposite ends of the network are the most divergent. In essence, MJ analysis provides a visually informative illustration of the relationships of a set of haplotypes segregating within a population.

As shown in Fig. 1d, MJ analysis of the *DAP1* haplotypes in European Americans indicates that risk HAP3 and protective HAP1 are at opposite ends of the network, indicating that these two regulatory alleles of *DAP1* differ by all 19 SLE-associated variants. As shown in Fig. 1e–f, MJ analysis of both the African American and Chinese populations found similar relationships for HAP1, HAP2, and HAP3 DAP1 regulatory haplotypes, although these haplotypes were at markedly different frequencies and related by more complex network intermediates. The most common risk haplotype was HAP3 in all three populations, which contributed significant SLE risk in European Americans (OR = 1.4, $p < 4.82e{-}03$), African Americans (OR = 1.7, $p < 0.05$), and Asians (OR = 1.6, $p < 0.04$). In addition, HAP1, although much less common in the AA and Chinese populations than in European Americans, consistently had decreased frequencies in SLE patients in comparison to healthy controls. As shown in Additional file 1: Table S7a, CMH analysis found that haplotype 3 (HAP3) was associated with risk for SLE in all three ethnic populations (OR = 1.5, $p < 4.51e{-}05$) and HAP1 was associated with protection from SLE in all three ethnic populations (OR = 0.7, $p < 1.88e{-}06$). Trans-ethnic meta-analysis of DAP1-SLE association using MR-MEGA program [49] further confirmed the observed association (Additional file 1: Table S7b). We also assessed the gene level dose effect of the risk allele in European American data set (Additional file 1: Table S8). We found that homozygotes for the risk haplotype have significantly higher risk (HAP3-HAP3, OR = 1.7) than those who carried just one allele either in combination with the protective allele (HAP1-HAP3, OR = 1.3) or in combination with HAP2 (HAP2-HAP3, OR = 1.5). However, due to the low frequency of the HAP3 risk haplotype, a larger sample size is required to achieve statistical significance of these associations. These findings indicate that the protective HAP1 and risk HAP3 alleles are highly divergent and have diametrically opposed effects on SLE susceptibility.

## SLE-associated regulatory haplotypes lead to reduced transcription of DAP1

A functional analysis of the 19 SLE-associated variants indicated that they all impacted transcription factor binding sites or are eQTLs affecting transcription of *DAP1* in PBMCs, lymphoblastoid cell lines (LCLs), monocyte-derived macrophages (MDMs), or monocyte-derived dendritic cell (MDDCs) panels (Additional file 2: Fig. S3a-c). Figure 2a presents the MJ network with just the three most common haplotypes within our European American cohort. As shown, the SLE protective allele HAP1 differs from the HAP3 risk allele by all 19 regulatory variants and from HAP2 by 15. As detailed in Additional file 1: Table S9a-9d, six of these regulatory variants, SNPs 1, 4, 8, 10, 13, and 16 have potent RegulomeDB scores, indicating that these variants may have a strong impact on transcription and/or chromatin structure. Figure 2b–d provides a snapshot of the UCSC genome browser across the genomic positions of SNP8 (rs2930047), SNP4 (rs267985), and SNP1 (rs3797111), illustrating transcription factors and/or accessory proteins that are likely to be affected by these SLE-associated regulatory variants. Further, as shown in Additional file 1: Table S10, all of the regulatory variant alleles embedded in the HAP3 risk allele are strongly associated with the downregulation of *DAP1* expression in published eQTL panels of human monocytes and PBMCs [52, 53]. SNP4 (rs2930047) shows the

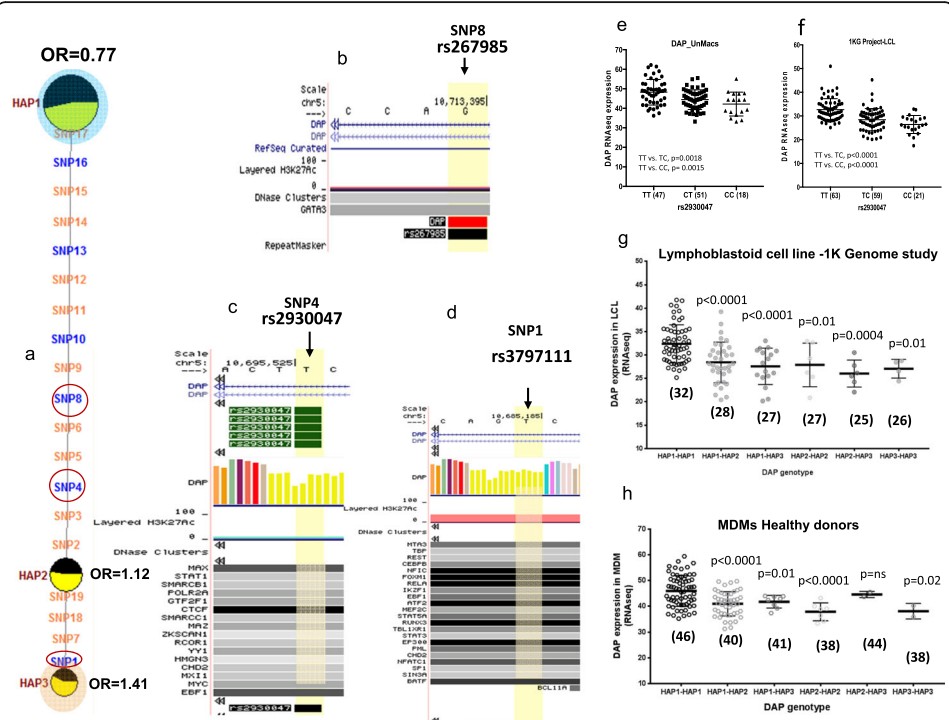

**Fig. 2** SLE-associated *DAP1* eQTLs in immune cells. **a** MJ network analysis on the top three most common *DAP1* haplotypes illustrates the position of six (blue color) potentially regulatory variants with strong RegulomeDB score and eQTL effects. MJ network nomenclature is the same as described earlier in Fig. 1. Encircled SNPs (SNP8, SNP4, and SNP1) indicate the variants with the strongest regulatory annotations based on ENCODE data and eQTL effects. **b–d** UCSC genome browser illustrates histone modification, transcription factor binding tracks based on ENCODE's ChIP-Seq data, and UCSC functional annotations for SNP8, SNP4, and SNP1. **e, f** *DAP1* eQTLs for rs2930047 SNP in human monocyte-derived macrophages (MDMs) from the present study and lymphoblastoid cell lines from the 1000 genomes study, respectively. **g, h** Effect of allele dosage on *DAP1* gene expression in different combinations of protective and risk alleles in lymphoblastoid cell line (LCL) samples from the 1000 Genome Project (**g**) and MDMs from the present study (**h**). Median value of *DAP1* expression in each diplotypes is shown underneath in parentheses

strongest eQTL effect in PBMCs ($Z$ score = − 16.4, $p$ = 5.16e−61, [54]) as well as in monocytes (naïve t-stat = − 9.4, $p$ = 4.58e−19).

In order to confirm the above eQTL effects of SLE-associated *DAP1* variants, we produced eQTL panels of MDMs, MDDCs, and ex vivo B cells of up to 116 healthy European American donors [25]. The transcriptomes of these cells were sequenced before and after stimulation with a TLR7/8 ligand (18-h stimulation), and each donor's genome was sequenced with the SLEv1 targeting array. In addition, the 1000 Genome Project (1KG Project) contains a panel of 220 lymphoblastoid cell lines (LCLs) that have both transcriptome and genome sequences available. As shown in Fig. 2e and f, *DAP1* expression is significantly ($p < 0.001$) reduced in MDMs of individuals with the SNP4 risk genotypes (*CT* or *CC*) as compared to those with protective genotype (*TT*) in both the unstimulated MDM panel and the 1KG Project LCLs, which is consistent with the published results with PBMCs and monocytes. Further, the detailed genomic sequence data available allows an assessment of HAP1, HAP2, and HAP3 *DAP1* regulatory alleles in specific genotypic combinations on the transcription of *DAP1*. As shown in Fig. 2 panels g and h, there is a progressive decrease in *DAP1* transcription as genotypes transition from HAP1/HAP1 homozygotes through various combinations into HAP3/HAP3 homozygotes. This decreasing spectrum of *DAP1* transcription is seen in both the 1KG panel of LCLs and our MDMs, although the frequencies of HAP2 and HAP3 homozygous genotypes are small in these limited collections (Fig. 2g, h). Furthermore, eQTL data in our EBV cell line panel, MDDCs and 1KG LCLs representing different world ethnic populations, all showed similar results (Additional file 2: Fig. S3c-f). We also studied *DAP1* expression in TLR7/8 stimulated MDMs and found the same trend of *DAP1* downregulation with the risk genotype, although overall *DAP1* transcription was reduced following TLR7/8 stimulation (Additional file 2: Fig. S3g). Taken together, these results indicate that a quantitative spectrum of *DAP1* transcription levels occurs in the human population with the HAP1 protective allele associated with the highest levels and the HAP3 SLE risk allele associated with reduced transcription in macrophages, dendritic cells, ex vivo B cells, LCLs, and PBMCs.

### Reduced DAP1 protein and increased autophagy phenotype in DAP1 SLE risk genotype

To study DAP1 protein level and their impact on autophagy in risk genotype, we purchased eight B cell-derived lymphoblastoid cell lines (LCLs) from the Coriell Institute (Additional file 1: Table S11). We also identified healthy volunteers with *DAP1* risk or protective genotype to perform experiments on primary cells, i.e., PBMCs and monocytes. As shown by Western blot analyses in Fig. 3a and b and Additional file 2: Fig. S4, DAP1 protein levels are significantly reduced in LCLs and in primary PBMCs and monocytes of healthy donors with the SLE risk (CC) allele in comparison to healthy donors with the SLE protective (TT) allele for SNP rs2930047. These results indicate that the regulatory polymorphisms that distinguish HAP1 and HAP3 result in a decrease of almost 40% in the level of DAP1 protein present in the immune cell lineages of individuals homozygous for the SLE risk allele of *DAP1*.

We next assessed the impact of these reduced levels of DAP1 on the magnitude of autophagy induction. Yahiro et al. have shown that DAP1 is a negative regulator of autophagy pathway [39]. Autophagy can be assessed by measurement of autophagic flux

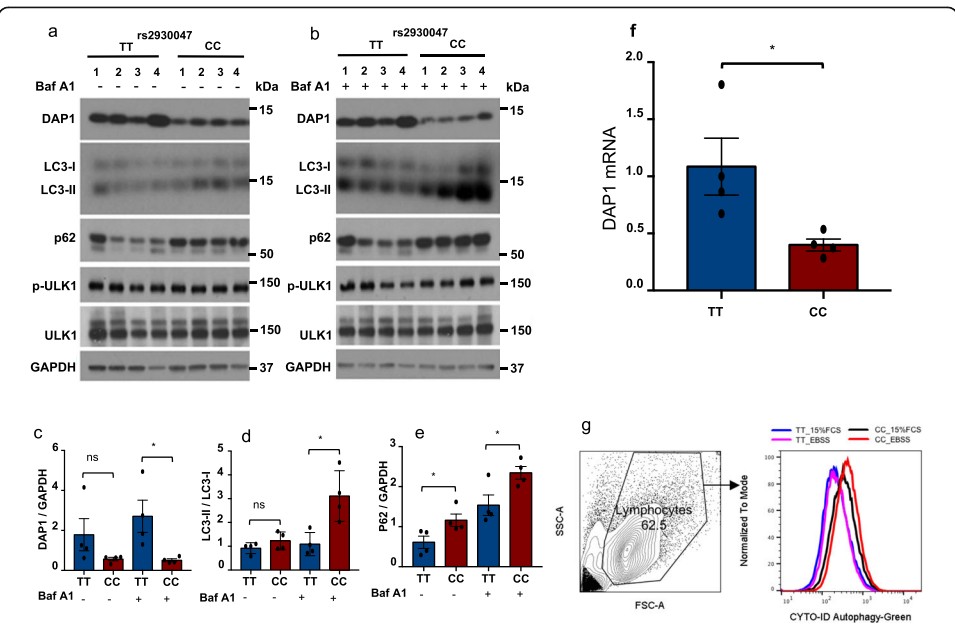

**Fig. 3** Reduced DAP1 protein and enhanced autophagy in DAP1 SLE risk genotype. B cell-derived lymphoblastoid cell lines (LCLs) on 8 participants from 1000 genome study were purchased from Coriell based on their *DAP1* genotypes. **a**, **b** Lymphoblastoid cells (LCLs) from four donors with protective genotype (TT) or four donors with risk genotype (CC) incubated with RPMI in the absence (**a**) or presence of 100 nM bafilomycin A1 (**b**) for 4 h. Cell lysates were analyzed by western blot using indicated antibodies. The ratio of DAP1/GAPDH (**c**) or the ratio of LC3-II/LC3-I (**d**) or the ratio of p62/GAPDH (**e**) is presented in bar graphs. Data are representative from at least three independent experiments. **f** Show quantitative RT-PCR analysis of *DAP1* mRNA in LCLs from four donors with protective genotype TT or four donors with risk genotype CC. *GAPDH* was used as an internal control to normalized *DAP1* expression. All plotted values are averages of two independent experiments using two different primer pairs. **g** Gating strategy and flow plots from CYTO-ID green autophagy detection assay on 4 TT or 4 CC LCL samples. Bar graph represents the average ± SEM of two independent experiments. Error bars: SEM; *$p < 0.05$. ns, not significant. Student's *t* test

that involves formation of autophagosomes and their fusion with lysosomes and their degradation [55, 56]. We assessed the impact of the *DAP1* polymorphisms on autophagy by comparing autophagy marker proteins between four LCLs homozygous for the SLE risk haplotype and four LCLs homozygous for the SLE protective haplotype (Fig. 3a). As shown in Fig. 3a, the levels of DAP1 protein are significantly reduced in LCLs with the SLE risk (CC) allele in comparison to LCLs with the SLE protective (TT) allele (Fig. 3a). Further, LCLs with the risk allele exhibited elevated ratios of LC3-II/LC3-I and expression of p62 at baseline or without bafilomycin A1 treatment (Fig. 3a), as well as in the presence of bafilomycin A1 (Fig. 3b), as quantified in Fig. 3c–e. Furthermore, *DAP1* transcription levels in eight LCLs were also confirmed via RT-qPCR (Fig. 3f). Results in LCLs were consistent with the observed enhanced autophagic flux in ex vivo PBMCs and monocytes of healthy donor with the SLE risk allele, indicating that basal as well as induced autophagic flux was enhanced in SLE risk (CC) allele (Additional file 2: Fig. S4).

In addition, a trend of enhanced autophagic flux was observed by flow cytometry both under nutrient-rich as well as under starvation conditions in LCLs of donors with the risk allele compared with those of LCLs with the protective allele (Fig. 3g). We also assessed mTOR kinase activity in these cells since previous studies have found that mTOR activity prevents Ulk1 activation by phosphorylating Ulk1 Ser 757 under nutrient sufficiency [57]. We observed that mTOR kinase activity in four LCLs with the

DAP1 protective allele is similar to that in four LCLs with the DAP1 risk allele, consistent with a previous report demonstrating that mTOR activity was not affected by DAP1 knockdown [42]. Overall, these results indicate that genetic variability in the transcription of DAP1 modulates baseline levels of the DAP1 protein in multiple immune cell lineages and that in turn impacts the level of autophagy in various immune cell lineages. However, future functional investigations on this pathway may provide full mechanistic insight.

### Transcriptome analysis in SLE patients with the *DAP1* risk haplotype

We assessed the downstream consequences of variations in *DAP1* transcription on gene expression profiles of PBMCs from 16 newly diagnosed SLE patients with either risk ($n = 10$) or protective ($n = 6$) *DAP1* alleles (Fig. 4a). As shown in the heatmap in Fig. 4b, cluster analysis identified 212 genes that were differentially ($p < 0.05$) expressed between SLE patients with the *DAP1* risk verses protective allele (Additional file 1: Table S12). As expected, *DAP1* transcription is significantly reduced in SLE patients carrying the *DAP1* risk allele (Fig. 4g), consistent with the eQTL studies described above and indicating that these patients should have enhanced autophagy activation in their immune cell lineages. Pathway analyses of the 212 divergent genes between SLE patients identified 11 upregulated (Fig. 4c) and 12 downregulated (Fig. 4d) biological pathways

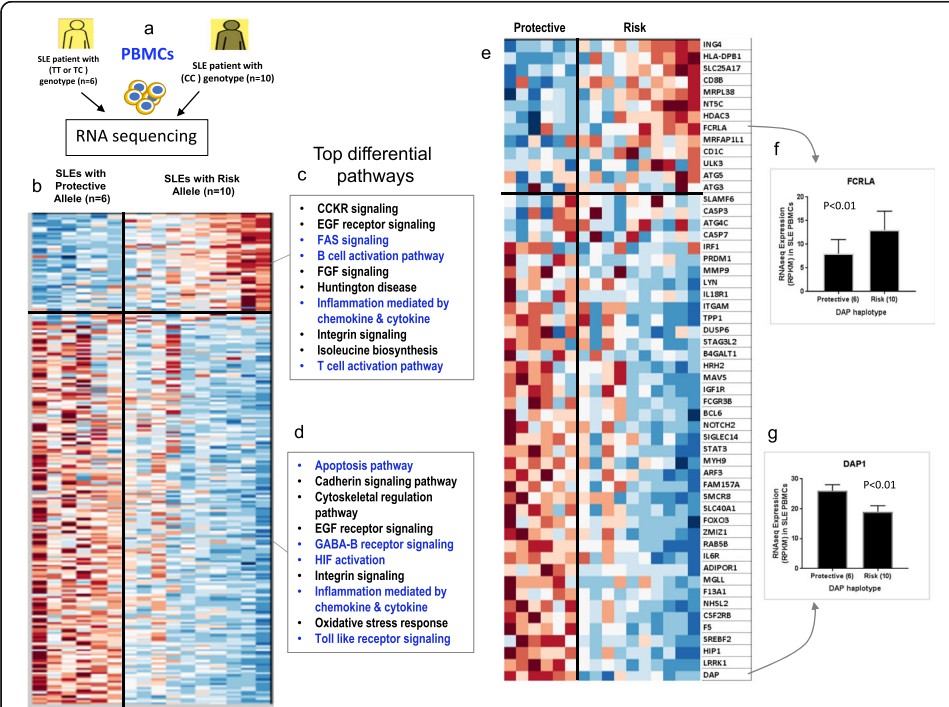

**Fig. 4** Gene expression analysis in PBMCs of SLE patients with and without *DAP1* risk. **a** Schematic for the RNA sequencing experiment in PBMCs of established SLE patients. RNA was extracted from PBMCs from SLE patients with rs2930047 protective genotype (TT, $n = 6$) or donor with rs2930047 risk genotype (CC, $n = 10$). **b** Heatmap shows the top two clusters of most differentially expressed 232 genes (RPKM values) between *DAP1* SLE risk and non-risk allele. **c, d** Top 10 biological pathways annotated for each cluster based on *PANTHER 14.1* classification (http://pantherdb.org). **e** Heatmap on 54 sub-selected genes to illustrate the status of some key biological molecules from the immune system, autophagy, and apoptosis pathways. **f, g** Expression plot of *FCRLA* and *DAP1* gene, as representative genes from each cluster

in PBMC from SLE patients carrying the risk allele (*PANTHER 14.1* classification (http://pantherdb.org)). Reactome pathway [58] analysis also identified 12 biological pathways including interleukin signaling, immune system signaling, cytokine signaling, apoptosis, and autophagy to be differentially regulated between *DAP1* risk and non-risk genotype (Additional file 1: Table S13). More detailed analysis showed that FAS signaling and both B and T lymphocyte activation pathways were upregulated in patients with the *DAP1* risk allele (Fig. 4c), whereas apoptosis, GABA B receptor signaling, TLR signaling, and several other pathways were downregulated in patients with the risk allele (Fig. 4d). Interestingly, several risk genes for autoimmune disease were among the top differentially expressed molecules. For example, expression levels of genes such as Integrin Subunit Alpha M (*ITGAM*) and Forkhead Box O3 (*FOXO3*), which have been previously shown to have decreased expression in SLE patients [11, 59, 60], were significantly further reduced in SLE patients carrying the *DAP1* risk allele compared to the protective allele. In this regard, mouse models of autoimmunity lacking *Itgam* have been shown to exhibit enhanced disease progression and inflammation [11, 61]. On the other hand, expression of *HLA-DPB1*, *CD8*, a regulator of T lymphocyte activation, and molecules expressed in activated B cells such as *FCRLA* (Fc Receptor Like A) were significantly upregulated in SLE patients with the *DAP1* risk allele in comparison to SLE patients with the protective allele (Fig. 4e, f). Previous studies have reported increased expression of these genes in PBMC from SLE patients in comparison to healthy controls [62, 63]. The heatmap in Fig. 4e presents some selected genes from Fig. 4b that are directly related to the immune system, autophagy, apoptosis, and other SLE implicated biological pathways. This gene expression analysis identifies some key differences in *DAP1* risk vs non-risk allele and identifies some genes that may drive B and T cell activation in SLE patients with the *DAP1* risk allele (Additional file 2: Fig. S5).

In summary, transcription analysis demonstrates that SLE patients stratified by *DAP1* genotype exhibit differential expression of several pathways associated with disease progression and severity in both animal models and human SLE. These results also suggest that stratification by *DAP1* genotype subsets SLE patients into separate groups with divergent patterns of immune cell activation.

### Autoantibodies associated with *DAP1* SLE risk haplotype

We measured the impact of *DAP1* risk alleles on humoral autoimmunity in a panel of 226 SLE patients. As shown in Fig. 5a, the production of anti-nuclear autoantibodies (ANA) in SLE patients with the *DAP1* risk haplotype was significantly ($p = 0.004$) higher than that of SLE patients with the protective genotype. We also performed high-throughput autoantigen array analysis in serum samples to assess the spectrum of antigens recognized by SLE patients with the *DAP1* risk allele using an autoantigen array containing 90 autoantigens previously implicated in a variety of autoimmune diseases including SLE [64]. As shown in Fig. 5b, the average normalized autoantibody levels were significantly ($p = 0.017$) greater in patients that were *CC* homozygotes than patients with the *TT* or *TC* protective genotypes. A comparison of autoantigen signatures in risk vs protective group showed significant (> 5-fold) enrichment for IgG antibodies to *Sm*, *SmD*, *U1-snRNP-BB*, and thyroglobulin (Fig. 5c, Additional file 1: Table S14a).

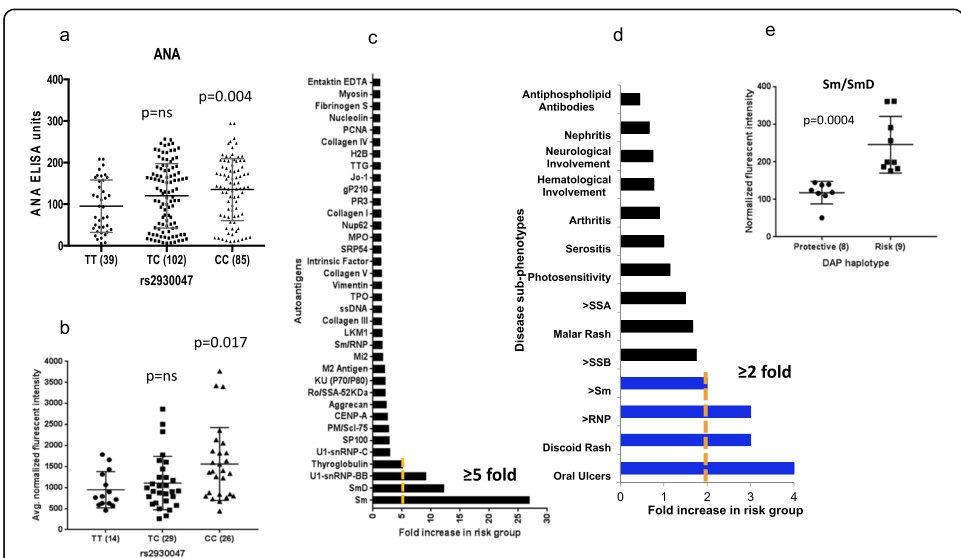

**Fig. 5** Autoantibodies in patients with the DAP1 risk haplotype. **a** Distribution of ANA in SLE patients with three different genotypes of the rs2930047 SNP. **b** Average normalized fluorescent intensity (NFI) for 89 autoantigens (IgG) in SLE patients with three different genotypes of the rs2930047 SNP. **c** Enrichment of specific autoantibodies in *DAP1* SLE risk haplotype compared to the protective haplotype. **d** Enrichment of specific autoantibodies and clinical phenotypes in *DAP1* SLE risk haplotype compared to the protective haplotype. In this independent set of samples, SLE patients with risk allele showed higher odds for clinical phenotypes such as oral ulcers, discoid rash and sm, and RNP antibodies. **e** Enrichment of sm/smD antibodies in *DAP1* SLE risk haplotype compared to the protective haplotype in a cohort of apparently healthy subjects that carried *DAP1* risk (*n* = 9) or protective (*n* = 8) genotype

To further validate this observation, we studied the association of carrying the *DAP1* risk allele with autoantibody specificity in two additional sets of SLE samples, one from China (Additional file 1: Table S14b; Additional file 2: Fig. S6) and another from Belgium (Fig. 5d, Additional file 1: Table S14c). As shown, we observed a consistent association of anti-sm and anti-snRNP antibodies with the *DAP1* risk allele cohort. Besides this, SLE patients that carried the *DAP1* risk allele showed higher odds for developing discoid rashes and oral ulcers than those who carried the non-risk allele (Fig. 5d). Furthermore, we genotyped a new cohort of 17 healthy volunteers for *DAP1* alleles and assessed their sm and RNP IgG titers in blood serum. We obtained very strong results as subjects with the *DAP1* risk allele showed significantly ($p = 0.0004$) higher sm/smD antibody titers than those with the protective allele (Fig. 5e, Additional file 1: Table S14d). However, quantitatively, levels of antibodies in healthy subjects were significantly lower than those in SLE patients. We also used cluster analysis on autoantigens to determine the spectrum of autoantigens recognized by patients with the *DAP1* risk allele. As shown in Additional file 2: Fig. S7, individuals with *DAP1* risk alleles preferentially produced autoantibodies against tightly clustered sets of non-nuclear antigens, while SLE patients with the *DAP1* protective allele showed no preferential antigen clusters.

Overall, these results suggest that subjects with the *DAP1* risk allele produce more autoantibodies and higher IgG responses that strongly favor the development of antibodies against distinct sets of antigens, i.e., sm and snRNPs, which may reflect differences in the source of autoantigen that initiates the breach in tolerance. Finally, clinical data analysis suggests that the effects of the *DAP1* risk allele may impact pathways that potentiate specific clinical disease features in SLE pathology.

## Discussion

These results demonstrate that *DAP1* regulatory polymorphisms modify autophagy levels in multiple immune cell lineages and that this increased autophagy significantly affects a variety of component phenotypes of SLE pathology. Figure 6 illustrates our current model for the impact of the *DAP1* risk allele on the progression and pathology of SLE. As shown, the HAP3 regulatory allele of *DAP1* mediates lower transcription of *DAP1* than the HAP1 regulatory allele, which results in a roughly 40% reduction in the autophagy-suppressing DAP1 protein. Our comparisons of the immune systems of SLE patients clustered by *DAP1* genotype revealed the consequences of this *DAP1* mediated endophenotype on the dysregulation of the immune system in SLE patients. Transcriptomic comparisons between newly diagnosed SLE patients clustered by *DAP1* genotype revealed several differences in their expression profiles for genes involved in the regulation of innate and adaptive immunity (*ITGAM*, *SLAMF6*), B and T cell activation (*FCRLA*, *CD8B*), the apoptotic pathway (*CASP8AP2*, *CASP7*), and autophagy (*ATG5*) (Fig. 4). These phenotypic variations were consistent with immune-system increased transcription of genes associated with autophagy and B and T cell activation (lymphocyte survival) and increased expression of HLA-D molecules associated with antigen processing and presentation (HLA-DRB, HLA-DQB, TAP2, etc.) (Additional file 2: Fig. S8). This is consistent with an extensive literature that associates enhanced levels of autophagy with SLE patients [34, 37, 65] with plasmablast development and persistence [35], T lymphocyte survival and proliferation [36, 66], and antigen presentation [40]. We have also demonstrated increased autoantibody titers and markedly increased autoantibody levels to autoantigens associated with severe pathology [67–69] and specific clinical disease manifestations including discoid rash and oral ulcers. Overall, our results illustrate the manner in which regulatory polymorphisms can manifest potent effects on susceptibility to autoimmune diseases such as SLE.

A key pathogenic consequence of decreased *DAP1* in SLE patients is the production of higher levels of autoantibodies against several autoantigens, notably including the Smith autoantigens. This observation was reproduced in two independent sets of

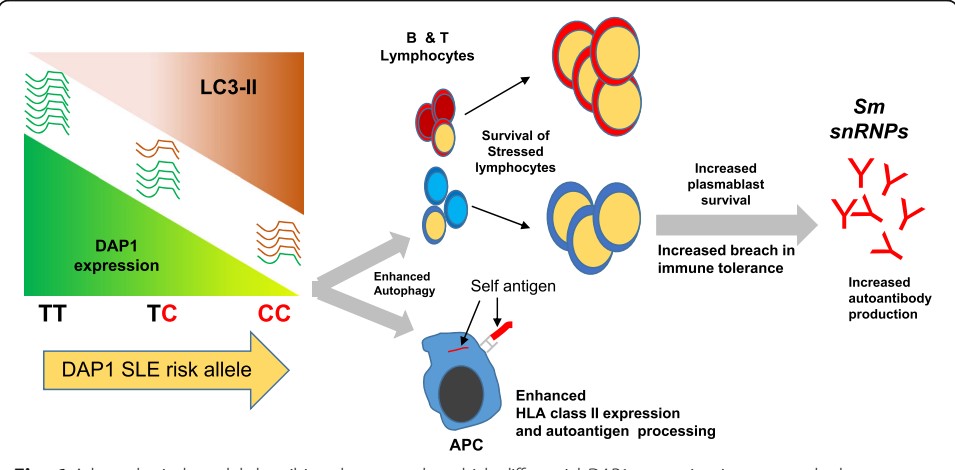

**Fig. 6** A hypothetical model describing the events by which differential *DAP1* expression impacts endophenotypes of autoimmunity and susceptibility to SLE. Our model proposes that the HAP3 regulatory allele of *DAP1* mediates lower transcription of autophagy-suppressing DAP1 protein, which results in enhanced autophagy, survival of autoreactive lymphocytes, enhanced antigen presentation, and development of autoimmunity

samples. Further, the enhanced production of Smith family autoantibodies is strongly associated with poorer prognosis in lupus nephritis [67], more severe disease, various clinical features [68], and earlier mortality [70]. The increased production of these auto-antibodies associated with the *DAP1* risk allele may be a major element in the associ-ation of this allele with SLE. The novel clustering of autoantigens recognized preferentially by the sera of SLE patients with the *DAP1* risk allele is also interesting and may indicate that increased autophagy preferentially promotes a breach in toler-ance to a specific source of autoantigen.

The ability of autophagy to decrease apoptosis in stressed cells and thus allow the in-creased survival of autoreactive lymphocytes in individuals with autoimmune disease is well-established and shown to be a common feature of the dysregulated immune sys-tem in many SLE patients [37]. Our analysis of transcriptional variations between SLE patients with and without the *DAP1* risk allele detected increased expression of *FCRLA*, *HLA-DPB1*, *CD1C*, *CD88*, and *PTPRCAP* (CD45) and decreased expression of *FOXO3*, *CASP1*, *CASP3*, *CASP7*, and *CASP8AP2*, which supports the hypothesis that low *DAP1* underlies increased autophagy, activated antigen presentation, B and T cell pathways, and decreased apoptosis of immune cell lineages in some SLE patients. Further, the en-hanced activities of the phagasome pathway in *DAP1* low expressing individuals indi-cate that antigen processing pathways are upregulated, consistent with the well-established impact of autophagy on the antigen processing pathways [40]. Taken to-gether, we hypothesize that increased autophagy throughout the immune system will enhance the survival of autoreactive lymphocytes, increase the presentation of endogen-ous self-antigens to autoreactive lymphocytes and the myeloid system, and thus in-crease autoantibody production and the prevalence and activation status of autoreactive cells. We admit that gene expression analysis in PBMCs is complex and limited in terms of specifically mapping the disease endophenotypes to specific cell type. It is therefore important in future studies to sort out the immune cells and per-form single-cell RNA sequencing (sc-RNA-Seq) analysis to figure out key gene expres-sion changes in specific cell types that directly regulate disease-associated pathways and promote autoimmunity. For example, although autophagy pathway was upregulated in the risk allele, it was not among the top differentially regulated pathways in the gene cluster analysis, which could be due to analysis on mixed populations of cells in the PBMCs. Therefore, future studies on single cell types might enrich the true signal and reveal the unknown.

The upregulation of autophagy in immune system cellular lineages is an established feature in SLE pathology [37]. Wu et al. studied the expression of autophagy-related molecules in PBMCs of SLE patients and found that the expression of several key mole-cules such as *Becline -1* and LC3 were correlated with SLEDAI score and autoantibody production [65]. Further, autophagy plays a crucial role in the long-term survival of B cells during activation [34]. Our data on primary B cells from healthy donors also con-firm reduced *DAP1* expression in risk genotype CC as compared to protective genotype TT (Additional file 2: Fig. S9). Consistent with the existing literature [71, 72], our LCL data on increased level of LC3-II and p62 in CC genotype clearly suggest enhanced au-tophagic potential of the SLE risk allele. Our data suggest that *DAP1* expression levels are a major element in the genetic underpinnings of these autophagy-mediated pheno-types. As with most disease phenotypes of SLE, there is significant qualitative and

quantitative variation among individual patients and our data indicate that *DAP1* regulatory polymorphisms establish the intrinsic limits of autophagy that will be induced in the immune cells of individual patients.

SLE develops through the combined effects of multiple factors and increased autophagy is only one of several interacting elements that impact disease severity and pathology [27]. There are autophagy-related genes that are associated with SLE [31, 32]. We confirmed the upregulation of many of these genes in SLE patients as compared to healthy subjects, and we demonstrate higher expression of several of these molecules in SLE patients that carried *DAP1* risk allele as compared to those with protective allele (Additional file 2: Fig.S10), suggesting potentially stronger autophagy induction of several disease-associated molecular pathways. Phenotypic diversity in SLE patient population makes disease management difficult and is likely to be responsible for the poor success rates of targeted drug therapies in SLE. Our analysis indicates that *DAP1* regulatory polymorphisms stratify SLE patient cohorts into subsets and that these subsets differ quantitatively in the components of disease pathology that they develop. This suggests that *DAP1* genotyping may be a useful strategy with which to explore variations in patient responses to drug therapies, especially for drugs that either target autophagy or pathways that are impacted by autophagy. Several drug trials are currently ongoing in SLE and other rheumatic diseases using drugs that either induce or inhibit autophagy [37].

DAP1 is a ubiquitously expressed, 15 kd cytoplasmic protein that contains a death-associated protein domain but lacks any known signaling motifs. The majority of the DAP1 protein is present in the cytoplasm in an inactive form due to phosphorylation by mTORC1 [42]. However, when mTORC1 kinase activity diminishes due to cellular stress, DAP1 is rapidly dephosphorylated and becomes active as a suppressor of autophagy. Thus, reduced mTORC1 kinase activity simultaneously induces autophagy through the ULK1 complex and activates DAP1 as a "brake" on the intensity of autophagy that will develop as the activation signal intensifies [41]. Although the precise molecular mechanisms by which DAP1 suppresses autophagy are unknown, DAP1 is hypothesized to decrease autophagy by competitively inhibiting signal transduction pathways that involve death domain motif interactions.

Our genomic analyses of *DAP1* diversity reveal that the regulatory enhancer elements for *DAP1* are remarkably polymorphic and that three functional, highly diversified alleles of this genomic segment are common in multiple ethnic groups. This functional polymorphism has several interesting genetic characteristics. First, these highly divergent regulatory alleles have persisted for an extended period as a stable polymorphism throughout the global expansion of man. Our analysis of transcriptional levels induced in various genotypic combinations indicates that the transcription rates dictated by these regulatory alleles are additive and lead to a spectrum of *DAP1* levels, spanning from very high levels in HAP1 homozygotes through various intermediate levels to very low levels in HAP3 homozygotes. Thus, *DAP1* regulatory polymorphisms create a continuous spectrum of variation in autophagy intensity within the global population. As a result, the levels of autophagy induced by cellular stress will be a continuous, quantitative variable within populations, which will promote flexibility in immune responses and in the development of adaptive memory within populations. Second, these three functional alleles are highly divergent and differ by many variations that impact

transcriptional/chromatin structure binding sites. Each regulatory allele has a specific constellation of regulatory polymorphisms that impact *DAP1* transcription by modulating the activity of multiple transcription factors in individual immune cell lineages. The specific combinations involve variations spanning the entire 73-kb LD block and the entire segment is in strong LD. In this regard, *DAP1* is expressed ubiquitously and it is likely that these alleles mediate similar effects throughout most cell lineages. However, it is quite feasible that these three common haplotypes have different functional effects on *DAP1* expression in cell lineages that are not in the immune system.

## Conclusion

Finally, these results demonstrate the efficacy and value of performing detailed, genomic sequence analyses to define regulatory alleles for key risk loci in complex diseases. The relevance of regulatory polymorphisms of *DAP1* to SLE was not effectively detected using the Immunochip and the relevant disease allele haplotypes could not be assembled from the SNP array assayed [18]. Our previous analyses of 28 SLE risk loci found a similar result for several risk loci, including the importance of the *XL9* segment in the *HLA-D* region [25]. The continued development of eQTL panels for specific cell lineages is critical to the efficacy of documenting and characterizing the regulatory effects of individual alleles. For example, trans-ethnic analysis shows a strong association of SNP1, SNP7, SNP18, and SNP19 with SLE, which are in the stem of the MJ network between HAP2 and HAP3 in all three ethnic groups. As shown in Additional file 2: Fig. S11, these variants impact transcription factor binding sites that are utilized by transcription factors strongly active in naïve B cells and other lymphocyte subsets. Further work on the functional effects of *DAP1* polymorphisms in B cell and T cell subsets may provide additional insights into the role of autophagy in driving specific elements of SLE pathology.

## Methods

### Study samples

This study was conducted on 1221 SLE patients and 811 healthy controls enrolled at six different research institutions (UT Southwestern Medical Center, Dallas, TX; Oklahoma Medical Research Foundation; Medical University of South Carolina; Feinstein Institute of Medical Research, New York; Université Catholique de Louvain, Brussels, Belgium; Xiangya Hospital, Central South University, Changsha, China) (Additional file 1: Table S15). All participants gave their written informed consent to participate in the research. All research protocols and experiment methods used in this study were approved by the UT Southwestern Institutional Review Board (IRB) under STU 042011-135 Dallas Regional Autoimmune Disease Registry Number, as well as by primary enrolling institution at each center. Clinical features for patients were obtained by chart review and collected using a standard data collection form. Clinical manifestations were evaluated according to criteria set by the American College of Rheumatology (ACR) [6, 7]. Subjects that exhibited four or more ACR criteria for clinical disease were classified as SLE patients by qualified rheumatologists. Statistical power analysis based on 2032 cases and control provided a power of 99.4% for SNP analysis and 90% for haplotype analysis (Additional file 2: Fig. S12). The source of serum/plasma and DNA/RNA

sample was peripheral blood. IRB-approved protocols were used to collect, transport, and store all the biological samples.

### Targeted sequencing of DAP1

Target enrichment and deep sequencing was carried out in the UT Southwestern Medical Center IIMT Genomics Core. About 1 μg picogreen measured genomic DNA was sonicated using *Covaris* S220 platform to generate 300–400-bp genomic fragments. The sequencing libraries were generated using TruSeq (Illumina) or KAPA Biosystem library preparation kits (KK8232). Each sample was ligated with custom-designed Illumina-compatible adaptors with unique 6 base barcodes following the kit manufacturer's protocol. The custom target enrichment array (Illumina Inc., San Diego, CA, www.illumina.com) was designed to capture the complete genome sequence of *DAP1* locus. The Illumina custom enrichment system theoretically captured sequence information for ~ 99.94% of the targeted region. The enriched libraries were sequenced on HiSeq2500 and HiSeq4000 platforms using a paired-end 100 bp protocol to produce high-quality sequencing data on each sample.

### Sequence alignment and variant calling

Sequence reads were demultiplexed, and each sample's reads were aligned to the human genome (HG19) using BWA-MEM, with base quality recalibration and local realignment performed with the Genome Analysis Toolkit (GATKv2) [73, 74]. As illustrated in Fig. 1a, target enrichment was highly specific and efficient, typically resulting in > 70% of reads on target and resulting in > 150X average coverage for *DAP1* locus. Coverage within the targeted segments was comprehensive with relatively uniform read depth throughout the non-repetitive regions. Samples with poor call rates (< 85%), poor sequencing fold coverage (< 25X), and significant $p$ value ($p > 0.001$) of HWE in controls were excluded from downstream analysis. Raw sequencing data (FASTQ files) for all targeted intervals in 1349 individuals is available on request (www.utsouthwestern.edu/labs/wakeland/about/contact.html).

### Defining high-quality samples and variants in studied populations

We used additional criteria to filter the SNP datasets to create an ethnically matched, case-control cohort with uniform coverage and high-quality variant calls. Samples with missing case/control status information and less than 85% call rate or duplicates were excluded from downstream analysis. Principal component analysis (PCA) was performed using standardized populations from the HAPMAP dataset as reference [75]. For the ethnic assignments, principal component analysis (PCA) was performed using HapMap reference data. Samples that clustered with HapMap CEU, YOR, and CHB groups were included in European, African American, and Asian study cohort, respectively. Application of these filters defined 1349 samples of European American ancestry, 363 samples of African American ancestry, and 320 samples of Asian ancestry for genetic analysis. Similarly, 2094 (in European American) and 1117 (African American and Asian) genetic variants of high quality were used for association analysis. Raw sequencing data (FASTQ files) for all targeted intervals in 2032 individuals is available on request (www.utsouthwestern.edu/labs/wakeland/about/contact.html).

### Variant annotation

Variant analysis with the GATK Haplotype caller identified 2447 QC pass variations in the targeted regions. Of these, 382 variants were shared among all three study groups (mostly about 70% were common variants), while 1712 variants were unique to the European cohort and 735 were unique to the African and Chinese cohort. Variants were annotated using multiple databases cataloguing the functional properties of human genomic variation, including the recent phase 3 release from the 1000 genome study, the PolyPhen/SIFT coding region database, the most recent release of the ENCODE database, and several recent expressed quantitative trait loci (eQTL) databases for immune cell lineages. The outcome of these analyses for all variants in the final dataset is summarized in Additional file 1:Table S2 & S3. Our sequence analyses identified an abundance of previously unannotated intronic and UTR3 variations in DAP1 gene. Functional annotation determined that 572/2094 (27%) of these variations were mapping to transcription factor binding sites according to ENCODE data (Additional file 2: Table S2). About 6% of the variations (135/2094) in the dataset were eQTLs for DAP1. Overall, about 32% (660/2094) variants were potentially functional (TF binding + DAP1 eQTLs), categorized as regulatory polymorphisms based on their localization into ENCODE-defined regulatory segments or inclusion in eQTL datasets.

### Genetic association tests and haplotype analysis

Of the 2094 genetic variants, 330 common allele frequency (MAF ≥ 0.05) markers were used for genetic association tests. A basic allelic association test was performed with 1349 PCA confirmed and age-matched European cases ($n = 773$) and controls ($n = 576$). The association test was controlled for genomic inflation using Golden Helix scripts where we first determined uncorrected genomic inflation factor ⋏ value which was 3.0. Similarly, African American and Chinese cohorts were also analyzed. We corrected data for batch effects and stratification with PCA using numeric association and regression analysis in Golden Helix. Finally, we corrected association results using this inflation value. This removed observed genomic inflation in association results. Chi-square $p$ values were further corrected for gender bias, using the covariate regression module in SVS, Golden Helix software. The LD structure and haplotype analysis in all three study cohorts was performed in SVS, Golden Helix, and using Haploview v4.2. Regulatory haplotypes were generated on 19 potentially functional variants with strong LD (≥ 0.8) to the study peak. An allele was defined as potentially functional if it is a coding variation, i.e., non-synonymous, synonymous, UTR or splice variant, and or an ENCODE's histone mark, transcription factor binding site, DNase I hypersensitivity clusters, or an expressed quantitative trait locus (eQTL). The same 19 markers were used to generate haplotypes in all the three ethnic groups, and results were compared.

### Production of monocyte-derived macrophages and dendritic cells

Human peripheral blood mononuclear cells (PBMCs) were obtained from adult healthy donors in accordance with the guidelines established by the Institutional Review Board (IRB) of the University of Texas Southwestern Medical Center (UTSW). These PBMCs were enriched by density gradient centrifugation of peripheral blood from healthy human donors through a Ficoll-Hypaque gradient. Monocytes were isolated from PBMCs

by negative selection using the EasySep™ Human Monocyte Isolation Kit (STEMCELL Technologies) according to the manufacturer's instructions. For the generation of monocyte-derived dendritic cells (MDDCs) or monocyte-derived macrophages (MDMs), monocytes were cultured in RPMI-1640 with 10% FBS, 2 mM L-glutamine, 10 mM HEPES, 1 mM sodium pyruvate, 100 U/ml penicillin, 100 μg/mL streptomycin supplemented with 100 ng/ml recombinant human GM-CSF and 50 ng/ml recombinant human IL-4 or 50 ng/ml recombinant human M-CSF, respectively. The culture media, which contained fresh cytokines, were replaced every 2 days, and MDDCs and MDMs were harvested on day 7. MDDCs or MDMs were seeded in 6 or 12 well plates at a density of $1 \times 10^6$ or $5 \times 10^5$ cells/well, respectively, treated and incubated with 10 μg/ml R848 for 18 h.

### Induction and assessment of autophagy in PBMCs and monocytes

PBMCs were incubated with EBSS (Thermo Fisher Scientific) in the presence of bafilomycin A1 (InvivoGen) for 4 h. Monocytes were incubated in RPMI-1640 medium with 10% FBS, 2 mM L-glutamine, 10 mM HEPES, 1 mM sodium pyruvate, 100 U/ml penicillin, 100 μg/mL streptomycin, and recombinant human M-CSF (50 ng/mL) for 14 h. Cells were lysed using CelLytic M Cell Lysis Reagent (Sigma-Aldrich). Ten micrograms of proteins was separated by SDS-PAGE and blotted onto nitrocellulose membranes. Blots were stained with antibodies against DAP1 (abcam), LC3 (Novus biologicals), or β-actin (Cell signaling), and staining was revealed with SuperSignal™ West Pico Chemiluminescent Substrate (Pierce). Protein densitometric analysis was performed with ImageJ software on scanned blots.

### Quantitative real-time PCR (RT-qPCR) and assessment of autophagy in LCLs

Five hundred nanograms of RNA was reverse transcribed using the SuperScript® III First-Strand Synthesis System (Invitrogen) according to the manufacturer's protocol. RT-qPCR was performed using PowerUp™ SYBR® Green Master Mix (applied biosystems) on the QuantStudio 7 Flex Real-Time PCR System (applied biosystems) according to the manufacturer's directions. Primers were as follows: GAPDH, 5′-tctctgccccctctgctg-3′ (forward) and 5′-agtccttccacgataccaaa-3′ (reverse); DAP1 primer pair 1, 5′-cccaagaacccagcacatc-3′ (forward) and 5′-gtaaggcaaaggacagagca-3′ (reverse); and DAP1 primer pair 2, 5′-caccaaa-gaagagaaagacaagg-3′ (forward) and 5′-gtggctgctggatgtgct-3′ (reverse). The comparative CT (ΔΔCT) method was used for data analysis. PBMCs or LCLs were incubated with EBSS (Thermo Fisher Scientific) or RPMI-1640 medium in the absence or presence of bafilomycin A1 (InvivoGen) for 4 h. Monocytes were incubated in RPMI-1640 medium with 10% FBS, 2 mM L-glutamine, 10 mM HEPES, 1 mM sodium pyruvate, 100 U/ml penicillin, 100 μg/mL streptomycin, and recombinant human M-CSF (50 ng/mL) for 14 h. Cells were lysed using CelLytic M Cell Lysis Reagent (Sigma-Aldrich). Ten micrograms of proteins was separated by SDS-PAGE and blotted onto nitrocellulose membranes. Blots were stained with antibodies against DAP1 (E59, Ab32056; 1:1000, abcam), LC3B (NB100-2220; 1:1000, Novus biologicals), P62 (NBP1-49954; 1:1000, Novus biologicals), ULK-1 (D8H5, 1:1000, Cell Signaling Technology), phospho-ULK1 (Ser757) (D7O6U, 1:1000, Cell Signaling Technology), β-actin (13E5, #4970; 1:1000, Cell Signaling Technology), GAPDH (14C10, 1:1000, Cell Signaling Technology), and HRP-conjugated anti-

rabbit (#7074; 1:1000, Cell Signaling Technology), and staining was revealed with Super Signal™ West Pico Chemiluminescent Substrate (Pierce). Protein densitometry analysis was performed with ImageJ software on scanned blots. We also assessed autophagy flux in SLE risk and non-risk genotype using flow cytometric assay [76, 77].

### RNA sequencing (RNA-seq) data production and analysis

RNA was extracted using TRIZOL (Life Technologies) and RNeasy Mini Kit (QIAGEN) according to the manufacturer's protocol. RNA quantity and purity was assessed on a NanoDrop 2000 spectrophotometer (Thermo Fisher Scientific), and integrity was measured on an Agilent Bioanalyzer 2100 (Agilent Technologies). RNA-seq libraries were prepared with the Illumina TruSeq RNA Sample Preparation kit (Illumina) according to the manufacturer's protocol. Libraries were assessed for insert size distribution and the presence of residual primer or primer dimers on an Agilent Bioanalyzer 2100. Sixteen RNA-seq libraries were sequenced on NextSeq550 flow cell using the SE75 protocol (single end 75 base pair reads), which yielded an average of about 22 Million reads/sample. We used CLC Genomics Workbench 7 for RNA sequencing and downstream bioinformatics and statistical analysis of the sequencing data. This approach used by CLC Genomics Workbench is based on a method developed by Mortazavi et al. [78]. Human Genome GRCh37 was used as reference sequence. The reference has 33,615 genes and 30,842 transcripts. All uniquely mapping reads to the genes were counted. Alignment with mismatch cost of "2," insertion cost "3," and deletion cost of "3" was used. The maximum number of hits for a read was set to 1 meaning that only reads those maps uniquely were considered. The steady-state expression of various genes was calculated in terms of RPKM values. For eQTL analysis, RPKM values were normalized as described previously [79, 80] as well as for population stratification or batch effect and cis-eQTL results were corrected for gender and ethnicity.

### Public databases used

Multiple public databases were used to validate and functionally annotate sequence variants identified in the present study. We used DNA and RNA sequencing data-based variants from the 1000 Genome Project samples (http://www.1000genomes.org/) and downloaded DNA sequencing data from the phase III dataset (ftp://ftp.1000genomes.ebi.ac.uk/vol1/ftp/release/20130502/) for haplotype analysis of 2504 genomic samples from the global human population. Similarly, FASTQ files of RNA sequencing data (http://www.geuvadis.org/web/geuvadis/RNA-seq-project) of lymphoblastoid cell lines derived from 369 Europeans were downloaded and used for DAP1 expression analysis. The ENCODE database (www.encodeproject.org) was used to annotate variants for transcription factors binding motif, DNase hypersensitivity cluster, and histone marks. Similarly, the RegulomeDB database (http://regulomedb.org/) was used to annotate potentially regulatory variations. Finally, UCSC genome browser (https://genome.ucsc.edu/) was used to generate custom tracks on sequencing variants and for the general visualization of study data.

### EBV cell line expression

Illumina mRNA sequencing libraries were prepared on 62 samples and sequenced on Illumina GAIIx/TruSeq 40-50 bp SE protocol. High-quality reads (> 20 million for each

sample) were assembled to the human genome (RefSeq transcript database, Build 37) using RNA-seq analysis application of CLC Genomics Workbench software. The expressions were normalized for gene length and measured as RPKM.

### ANA ELISA in SLEs

ANA ELISA was performed in 226 SLE sera/plasma samples to detect the expression of antibodies to chromatin (dsDNA and histones), Sm/RNP, SS-A, SS-B, Scl-70, centromere, PCNA, Jo-1, mitochondria (M-2), and ribosomal-P protein in serum/plasma samples using kits (Cat. No. 708750) from Inova Diagnostics Inc., San Diego, CA.

### Autoantigen protein array analysis in SLE

Serum/plasma samples from 69 SLE patients and healthy individuals were run on an auto-antigen protein array to identify the antigen specificities of autoantibodies observed in each group. The array used in the present study screened > 80 antigens previously implicated in a variety of autoimmune diseases including SLE. Autoantigen microarrays were manufactured, hybridized, and scanned as described previously [64]. Briefly, antigens were diluted to printing concentration in printing buffer (Whatman, Sanford, ME, USA) and transferred to 384-well plates. The antigens were printed in duplicate onto nitrocellulose-coated 16-pad FAST™ slides (Whatman, Sanford, ME, USA) using a MicroGrid II 610 bio-robotics array printer (Genomic Solutions Inc., Ann Arbor, MI, USA). After printing, the slides were kept in a chamber with 70% humidity for 4 h at room temperature and stored at 4 °C. For hybridization, slides are warmed to room temperature and 60 μl blocking buffer (Whatman, Sanford, ME, USA) is added to each array for 60 min. Serum samples are pretreated with DNAse-I (50 U/ml) for 30 min at room temperature in buffer containing 50 mM Tris-HCl, 75 mM KCl, 3 mM MgCl2, and pH 8.3. The pretreated serum samples are diluted 1:50 in blocking buffer and then added to each array for 1 h. After that, each slide array is washed with 100 μl of washing buffer (Whatman, Sanford, ME, USA) three times for 5 min each, and then Cy3-conjugated anti-human IgG and Cy5-conjugated anti-human IgM (Jackson Immuno Research, West Grove, PA, USA) at 1:1000 dilution are applied to each array incubated at room temperature for 1 h. Genepix 4000B scanner with 532-nm and 635-nm wavelengths and Genepix Pro6.0 software are used to generate the Gene Pix Results (GPR) files. The net fluorescence intensities (nfi) are normalized using anti-human IgG/IgM on each array as previously described [64]. Heat maps were generated using Cluster and Treeview software (http://rana.bl.gov/EisenSoftware.htm). The details and sources of these antigens are available on request.

### Statistical analyses

We used SNP & Variation suite (SVS) of Golden Helix (version 7.6.8 win64, *Golden Helix, Inc., Bozeman, MT,* www.goldenhelix.com) for genetic analysis. SNP conditioning analysis was performed using the regression module of SVS. Haploview v.2 software was used for visualization of LD plots and haplotype analysis [81]. GraphPad Prism 6.0 software was used for statistical analysis and graphics. Correlations between continuous variables were determined using Pearson's *r* in GraphPad Prism 6.0. Discontinuous variables were compared by Fisher's exact test. *p* values < 0.05 were considered significant.

Trans-ethnic meta-analysis was performed using MR-MEGA Program [49, 50].

### Gene expression data analysis methods

Methods for data normalization and analysis are based on the use of "internal standards" that characterize some aspects of the system's behavior, such as technical variability, as presented elsewhere [79, 80, 82]. Created initially for the analysis of microarray data they were slightly modified to the needs of RNA-seq data analysis.

Differential gene expression analysis was carried out using methods described in earlier publications [79, 80, 82]. These include the following steps:

1. Construction of the "reference group" by identifying a group of genes expressed above background with inherently low variability as determined by an $F$ test. The "reference group" presents an internal standard of equal expression. As such, the "reference group" is used to assess the inherent variability resulting from technical factors alone (technological variation). By creating an estimate of the technological variation, we are able to select a group of biologically stable genes.

2. Selection of replicates using the commonly accepted significance threshold of $p < 0.05$ with a Student $T$ test. This selection maintains the commonly accepted sensitivity level; however, a significant proportion of genes identified as differentially expressed at this threshold will represent false positive determinations.

3. An associative $T$ test in which the replicated residuals for each gene from the experimental group are compared with the entire set of residuals from the reference group defined above. The $Ho$ hypothesis is checked to determine whether the levels of gene expression in the experimental group presented as replicated residuals (deviations from the averaged control group profile) is associated with a highly representative (several hundred members) normally distributed set of residuals of gene expression values in the reference group. The significance threshold is then corrected to render the appearance of false positive determinations improbable: in the current case $p < 0.0001$. Only genes that pass both tests are presented in the final selections. Additional restrictions were applied to the minimal gene expression level (RPKM > 2) and fold of changes (> 1.5). The two-step normalization procedure and the associative analysis functions are implemented in MatLab (Mathworks, MA). Functional analysis of identified genes was performed with Ingenuity Pathway Analysis (IPA; Ingenuity® Systems, Redwood City, CA, http://www.ingenuity.com).

## Supplementary information

---

**Additional file 1: Table S1.** DAP1-SLE association results based on Immunochipv.1. **Table S2.** List of sequencing variants in DAP1 gene region-European cohort. **Table S3.** 19 SLE associated DAP1 variants used for haplotype analysis. **Table S4.** List of variants in African-American and Chinese cohort. **Table S5.** Cochran-Mantel-Haenszel test result on SNPs. **Table S6.** DAP1 haplotype association analysis results. **Table S7.** Trans-ethnic analysis by Cochran-Mantel-Haenszel and MR-MEGA program. **Table S8.** Diplotypes analysis in Caucasian data. **Table S9.** Potentially regulatory variant annotations. **Table S10**- eQTL effects of SLE associated variants. **Table S11.** Lymphoblastoid Cell Lines (LCL) used in study. **Table S12.** PBMCs RNA-Seq analysis. **Table S13.** Reactome pathway analysis. **Table S14.** Autoantibody data. **Table S15.** List of participating institutions and investigators.

---

**Additional file 2: Fig. S1.** DAP1 genetic association with SLE. **Fig. S2.** Genomic position of SLE associated DAP1 variants. **Fig. S3.** DAP1 eQTL with rs2930047. **Fig. S4.** Autophagy induction data in PBMCs and monocytes. **Fig. S5.** Gene expression data. **Fig. S6** Sm and *snRNP* antibodies in DAP1 SLE risk allele. **Fig. S7.** Heatmap of non-nuclear autoantigen signatures. **Fig. S8** Expression of antigen presentation pathway**. Fig. S9** DAP1 expression in primary B cells. **Fig. S10** RNA-Seq data in PBMCs. **Fig. S11.** HAP3 defining key variants. **Fig. S12.** Statistical power analysis. **Fig. S13.** Uncropped blots.

**Additional file 3.** Review history

### Acknowledgements
The authors are very thankful to SLE patients and healthy volunteers whose contribution was essential to conduct this study. We are grateful to the personnel in the Genomics Core at UT Southwestern Medical Center and other collaborating laboratories and centers for their excellent technical support and participation in the study.

### Review history
The review history is available as Additional file 3.

### Peer review information

### Authors' contributions
P.R. performed the experiments, genetic analysis, and manuscript writing; R.S. performed the experiments, analyzed the data, and wrote the manuscript. H.Z. contributed the samples; L. R. contributed to the experiments and reagent procurement; D.J. and Y.G. performed the experiments; C.L., C.A., B.Z., J.Z., J.A.K., and J.M.G. contributed to the sample collection, processing, and sequencing. B.E.W. and I.D. performed the data analysis. B.R.L., N.J.O., G. G., B.P.T., P.M.G., P.K.G., J.A.J., X. Z., and D.R.K provided the samples. S.K.N., C.P., N.V.O., and Q.Z.L contributed to the analysis and manuscript editing. E.K.W. conceived and designed the experiments and edited the manuscript. The authors read and approved the final manuscript.

### Funding
The authors gratefully acknowledge the funding from Alliance for Lupus Research (ALR) and grants that supported E.K.W.: RC2-AR-058959, R37AI04519607, P50-AR05550; J.A.J.: P30AR073750, U19AI082714, R01AR072401; B.P.T.: AR071410, AR071947; D.R.K: P50 AR055503; and S.K.N.: AR060366. Funding from the Walter M. and Helen D. Bader Center for Research on Arthritis and Autoimmune Diseases is also acknowledged.

### Availability of data and materials
All raw sequencing data are available on NCBI SRA with IDs PRJNA588226 [83] and PRJNA668797 [84]. Details about sequencing variants, source files, and their annotations are provided in the supplementary information of the manuscript.

### Ethics approval and consent to participate
All participants gave their written informed consent to participate in the research under an approved IRB # STU 042011-135 of Dallas Regional Autoimmune Disease Registry. All research protocols and experiment methods used in this study were approved by the UT Southwestern Institutional Review Board (IRB) as well as by primary enrolling institution at each center. All experimental methods comply with the Helsinki Declaration.

### Competing interests
The authors declare no competing interests.

### Author details
[1]Department of Immunology, University of Texas Southwestern Medical Center, Dallas, TX 75390, USA. [2]Department of Rheumatology and Immunology, Xiangya Hospital, Central South University, Changsha 410008, China. [3]Department of Molecular Genetics, University of Texas Southwestern Medical Center, Dallas, TX 75390, USA. [4]Arthritis and Clinical Immunology Program, Oklahoma Medical Research Foundation, Oklahoma City, OK 73104, USA. [5]Institut de Recherche Expérimentale et Clinique, Université catholique de Louvain, 1200 Bruxelles, Belgium. [6]Division of Rheumatology, Department of Medicine, Penn State Medical School, State College, PA, USA. [7]Division of Rheumatology and Immunology, Medical University of South Carolina, Charleston, SC, USA. [8]Feinstein Institute of Medical Research in Manhasset, New York, USA. [9]Rheumatic Diseases Division, University of Texas Southwestern Medical Center, Dallas, TX 75390, USA.

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

## 
