## [**Additional file 3.** Review history · Genome Biology]

Review history

First round of review

Reviewer 1

Raj et al performed deep sequencing of the DAP1 genetic locus to evaluate association with SLE risk, in a modest multi-ethnic cohort of SLE patients and controls. A low frequency risk haplotype was identified, which is correlated with decreased expression of DAP1 mRNA and protein and with other clinical characteristics like autoantibody production.

More details are needed to justify and understand the study design. It seems that DAP1 was resequenced as part of a previous, larger study that aimed at fine mapping candidate SLE associated loci, but it is not clear if results presented in the current manuscript were already described in the previous study. Previous genetic evidence of association with SLE was weak for this locus, what was the rationale to prioritise this region over others? Also, there is no detail on how the DAP1 region was defined and what are coordinates of this resequenced region.

Methods: information about how many patients and controls came from the different research institutions is lacking. Clinical characteristics of patients should also be described; authors state in the manuscript that SLE is clinically heterogeneous, how heterogeneous was the cohort, and could this affect results?

What was the statistical power of the study?

Figure 1b shows LD patterns for a region that is larger than the targeted re-sequenced DAP1 locus; where is the rest of the data shown coming from?

What population was used to plot the LD patterns shown in Fig1C? rs267985 is the top associated SNP in Caucasians only.

Meta-analysis: Recently, more sophisticated analysis methods have been developed for trans-ethnic fine mapping (for example, MR-MEGA in Magi et al Human Molecular Genetics 2017 and Morris et al Nature Communications 2019). Was this considered for the analysis of the data presented in the manuscript?

Were the rare variants identified in the re-sequencing study analysed for association with SLE?

The authors claim that the risk SNP is correlated with decreased expression of DAP1 in B cells, but this is not supported by the results found on primary B cells shown on Supplementary Figure 3, this should be discussed.

Why was the transcriptome analysis performed in a heterogeneous cell population such as PBMCs? What was the individual cell types distribution in patients with the risk alleles compared to the patients with the protective alleles? Are the p value reported for differentially expressed genes corrected for multiple testing?

Autophagy was not one of the top differential pathways in this analysis (figure 4); did the authors expect to see a difference?

Minor comment:

Gene names should be written in italics and protein names in non-italics.

Reviewer 2

Authors conducted a deep sequencing study of the SLE case-control cohort, in a target gene approach. They incorporated ~2,000 SLE cases and controls. The target gene was DAP1. Authors identified suggestive associations of the DAP1 common variants on SLE risk. Expression quantitative trait (eQTL) effects.

1. This is not clear for the reviewer why the authors only focused on the DAP1 genes. Since a variety of

SLE genes have been reported, there needs rationale to select DAP1, which this reviewer could not find.

2. The merit of deep sequencing is not clear. The identified variants were common, and the typical GWAS and subsequent imputation can capture them, so no need to conduct deep-sequencing.

3. SLE risk variants are likely to have eQTL effects of white blood cells. So the findings were not novel.

4. Significance of the identified association signals were not enough. Considering potential multiple comparisons on other genes, at least, the typical genome-wide significance threshold is required ($P < 5 \times 10^{-8}$).

Reviewer 3

Following up on nominal association of rare genetic variants in an SLE ImmunoChIP study, the author performs deep sequencing of the DNA surrounding the DAP1 gene in patients with SLE and controls of various ancestries. After identifying and replicating association of a rare haplotype not covered by the ImmunoChIP, the author identifies a robust eQTL (at mRNA and protein levels) and allelic autoantibody association. The genetic and biological conclusions of this study were replicated in independent cohorts of patients, which provides confidence in the significance and impact of this study. Despite reviewer enthusiasm for the findings of this manuscript, the format and lack of clarity in the supplemental materials was problematic. Altogether, this is an exciting manuscript that will benefit from extensive reformatting of the Tables and Supplements.

Major comments:

- * The authors identify a large variance in DAP1 expression levels that is in part explained by the SLE-risk haplotype. Is there a genotype-dependent change in the variance in DAP1 expression (as assessed using variance association mapping)? This analysis would also be particularly interesting in the autoantibody associations presented in Figure 3.
- * In the introduction, the authors mention that a variant in DAP1 is associated with IBD at genome-wide significance, what is the exact relationship of the IBD-associated variant to the SLE association in terms of linkage disequilibrium?
- * The authors report that 27% of variants in DAP1 mapped to known ENCODE transcription factor binding sites. Are any predicted to break or change the preferred TF binding site?
- * What multiple testing correction was applied in the RNA-seq analysis presented in Sup Table 9? The term "average expression protection (6)" is not clear.

Minor comments (these are minor because they can be very easily addressed, but the formatting issues are extremely important for manuscript clarity and usefulness).

- * Sequencing of the DAP1 gene identified numerous rare and common variants, and the authors indicate that DAP1 is an extremely polymorphic gene. Is this region of the genome more polymorphic (statistically) than other regions encoding protein-coding genes (when accounting for gene length)?
- * European/European American is preferable to Caucasian as the African American equivalent to Caucasian is socially insensitive.
- * The authors should use rsIDs or genomic position (chr:position) rather than new identifiers (such as SNP1, SNP2, SNP3) to allow subsequent studies to easily identify and assess these variants in their datasets.
- * As presented in the Reviewer pdf, Table 1 was extremely hard to read and interpret (it spans several pages). RegulomeDB annotations need to be defined in a Table legend. It was difficult to identify the reason some rows were highlighted in yellow.
- * In Table 2, the table also spans multiple pages and is hard to review. The column rows need to indicate that a haplotype frequency is given. The ChiSquared p-value should be limited to three significant digits, and the OR column should be formatted as to not give an OR for non-significant associations (e.g. the OR of 5.2 is not significant given a p-value of 0.08; similarly, if both the cases and controls lack a haplotype, the OR cannot equal 100).
- * It would be helpful to add a supplemental figure showing the gene DAP1 in the context of the genetic variants associated with SLE.
- * Supplemental Table 2a is unhelpful as presented - the reviewer was unable to assess this table in

the current format. This should be a supplemental dataset that can be provided as a tab-delimited txt file or excel file.

- * Supplemental Table 5 was hard to interpret as it also spanned two pages. Additionally, it was confusing way the ORs of specific Diplotypes were highlighted as they had non-significant p-values (p -values >0.05)
- * Supplemental Table 6 should also be reformatted as a txt file.
- * The motifs on page 481 are not formatted in an interpretable fashion.
- * Supplemental Table 7 needs to be reformatted - either with smaller txt or as a txt file.
- * Supplemental Table 10 needs to be reformatted - the table can be smaller to fit on one page and the legend needs to be formatted to be legible.
- * In the reviewer pdf, Supplemental Table 11 does not have a legible title.
- * Supplemental Table 11 should be a txt file. The ratio and p-value columns should show three significant digits. A README tab should clearly identify the source of values provided.

Authors' response to reviewers

Deep sequencing in 1200 SLE patients reveals DAP1 regulatory haplotype that potentiates autoimmunity across ethnicities.

We appreciate the time and efforts of the reviewers in evaluating our manuscript and providing constructive suggestions to further improve it. We respond to their comments and queries and have provided additional analyses and data as they have suggested. Segments in the text either re-written or updated have been inserted into the revised manuscript. We also include additional data and information in the response to the reviewers questions.

Reviewer #1:

Q1. Raj et al performed deep sequencing of the DAP1 genetic locus to evaluate association with SLE risk, in a modest multi-ethnic cohort of SLE patients and controls. A low frequency risk haplotype was identified, which is correlated with decreased expression of DAP1 mRNA and protein and with other clinical characteristics like autoantibody production.

More details are needed to justify and understand the study design. It seems that DAP1 was resequenced as part of a previous, larger study that aimed at fine mapping candidate SLE associated loci, but it is not clear if results presented in the current manuscript were already described in the previous study. Previous genetic evidence of association with SLE was weak for this locus, what was the rationale to prioritise this region over others? Also, there is no detail on how the DAP1 region was defined and what are coordinates of this resequenced region.

Response: Our previously published study was focused on common risk alleles and on variants with minor allele frequency $\geq 5\%$ in study population. Rare and low frequency genetic variants were not analyzed. DAP1 locus was not investigated in this previous study (1).

Why we targeted this locus? The DAP1 locus was the focus of our study since it recently suggested to be a candidate gene in a familial linkage study of SLE and RA (2, 3). Also, meta-analyses of GWAS datasets had associated variations in DAP1 with ulcerative colitis and IBD (4, 5). Given our extension data set and large population cohort, we decided to directly address the association of this locus with autoimmunity.

it is true that previous investigations, based on genotyping arrays such as ImmunoChip, observed a relatively weak signal of association at DAP1 and disease. We hypothesized that the poor association was due an absence of enough functional variants on these arrays. For example, SNP arrays only capture a few of the common frequency variants or tag SNPs within a given locus. There were only 12 markers from entire DAP1 gene captured on ImmunoChipv.1 (See list in Suppl. Table.1), and consequently, the peak association with SLE with so few markers resulted in DAP1 being a marginal candidate gene. Much stronger data concerning the functional variants of DAP1 that drive the association statistics was needed. As with all the other SLE risk loci that we have analyzed by population sequencing, much stronger associations are detected by sequencing variants as opposed to SNP tagging

available in the ImmunoChip or other GWAS SNP arrays.

For these reasons, we selected DAP1 for targeted deep sequencing, aiming to capture all common and rare genetic variants at this locus that may underlie the previously reported weak association with SLE. We sequenced 102 Kb of a genomic segment exhibiting a strong LD (the LD block that contained these DAP1 associated SNPs) in 1221 SLE patients and 811 healthy controls in three ethnic groups. The following table shows the genomic coordinates of tiled DAP1 region.

Chromosome	Start pos hg19	End position hg19	Gene	Tiled region size	Fold coverage
5	10678000	10780000	DAP1	102,001 bp	154.7

Q2. Methods: information about how many patients and controls came from the different research institutions is lacking. Clinical characteristics of patients should also be described; authors state in the manuscript that SLE is clinically heterogeneous, how heterogeneous was the cohort, and could this affect results? What was the statistical power of the study?

Response: As per reviewer's suggestions, Suppl. Table 12 summarizing number of samples contributed by various institutions has been provided in revised manuscript.

Enrolled patients were having minimum of 4 or more criteria for SLE classification according to American College of Rheumatology. A large body of literature has established that a huge clinical heterogeneity exists in SLE patient population. However, in the present study, we did not stratify patients based on their clinical characteristics for the genetic association analyses. For our current study, established SLE patients were treated as cases and non-SLE healthy volunteers were considered as controls for genetic association analysis. Our results clearly replicate genetic association with SLE patients in multiple ethnic groups, however, it is possible that a clinical sub-category of SLE may show stronger association than overall cohort. Despite the fact that our study deals with a low frequency haplotype which is present in 10-12% frequency in population, Cochran-Mantel-Haenszel test (across three ethnic groups) P-value reached as low as $3.E-05$ and 0.0013 for peak variant and risk haplotype, respectively, in about 1200 study samples.

Statistical power of 80% or greater is widely used to avoid false negative genetic associations and to determine a correct sample size in genetic association studies. We obtained a power of 99.4% for SNP analysis and 90% for haplotype analysis, see graphs below, which is strong.

Figure 1b shows LD patterns for a region that is larger than the targeted re-sequenced DAP1 locus; where is the rest of the data shown coming from?

Response: The Linkage Disequilibrium (LD) block shown in Figure 1b is generated based on Caucasian samples from 1KG genome study (<https://www.internationalgenome.org>). The purpose of this data was to show the LD structure in and around DAP1 locus. As shown in this figure, LD is strong in the region containing DAP1 locus but it does not extend much into flanking regions.

What population was used to plot the LD patterns shown in Fig1C? rs267985 is the top associated SNP in Caucasians only.

Response: The Linkage Disequilibrium (LD) block shown in Figure 1c is generated based on Caucasian SLE and control samples from present study. rs267985 is the peak variant in Caucasians only. But this is not surprising given that frequency of DAP1 variants may vary between ethnic groups and also, different number of samples were analyzed from African and Asian population. Most interesting finding is SLE risk haplotype that include this variant and consistently pose significant risk to SLE across ethnicities.

Meta-analysis: Recently, more sophisticated analysis methods have been developed for trans-ethnic fine mapping (for example, MR-MEGA in Magi et al Human Molecular Genetics 2017 and Morris et al Nature Communications 2019). Was this considered for the analysis of the data presented in the manuscript?

Response: We agree with reviewer. We had not applied this method to our data set earlier. But now as per reviewer's suggestions, we did perform meta-analysis on our top 19 DAP1 SNPs using MR-MEGA program. The results from MR-MEGA further support our original findings (Supplementary Table 4C). Many markers included in SLE risk haplotype show strong p-value of association with SLE in meta-analysis.

See snap shot of the table below. This method has been cited in the revised manuscript.

Were the rare variants identified in the re-sequencing study analysed for association with SLE?

Response: Yes. Low frequency variant association analysis showed the association. In addition, we also applied KBAC (the kernel-based adaptive cluster) method that uses counts of multi-marker genotypes to perform a special multivariate case/control test to determine their association with the phenotype. This test p value is 0.035 for DAP genotypes in SLE. See summary results in following table:

Gene Name	P-Value (One-Sided)	-log10 P-Value (One-Sided)	KBAC (One-Sided)	Bonf. P (One-Sided)	FDR (One-Sided)	Sample Size Used	# Markers	# Multi-Marker Genotypes
DAP	0.035914086	1.065935626	0.291620188	1	0.372294372	1349	478	544

The authors claim that the risk SNP is correlated with decreased expression of DAP1 in B cells, but this is not supported by the results found on primary B cells shown on Supplementary Figure 3, this should be discussed.

Response: Due to the small number of primary samples available for this analysis, statistical significance was not obtained. The data still support the expected trend of reduced DAP expression in risk allele. To further confirm this observation and support our primary cell data, we purchased B cell derived lymphoblastoid cell lines (LCLs) from Coriell (<https://www.coriell.org>) and performed western blot analysis, which demonstrated significant reduction in DAP1 expression in risk allele (see Figure 3A,B,C & F). Also, RNA sequencing data on LCLs confirmed reduction of DAP1 transcription in risk genotype (Figure 2G).

Fig.3 Fig.2

Why was the transcriptome analysis performed in a heterogeneous cell population such as PBMCs? What was the individual cell types distribution in patients with the risk alleles compared to the patients with the protective alleles? Are the p value reported for differentially expressed genes corrected for multiple testing?

Autophagy was not one of the top differential pathways in this analysis (figure 4); did the authors expect to see a difference?

Response: PBMCs are most commonly used sample type in human gene expression studies, so we investigated the DAP1 expression in this cell type. It is complex to precisely normalize and compare alleles for their cell type composition and proportion. This would vary significantly in both healthy population as well as in patients. We believe it would be gene expression of key genes in disease associated biological pathways that regulate mechanisms and pose risk or provide protection to an individual. Yes, p-values were corrected for multiple testing. Autophagy pathways did not show up among top differentially regulated pathways in the gene cluster analysis, but this simply be due to the mixed populations of cells in the PBMCs. We did see an upregulation of FAS signaling in the risk genotype group, which has been reported to enhance both autophagy and apoptosis. Autophagy pathway was also upregulated in risk allele but it was not among the top 10 presented in the figure.

Minor comment:

Gene names should be written in italics and protein names in non-italics.

Response: We have corrected gene names to italics and protein names to non-italics in revised manuscript.

Reviewer #2:

Authors conducted a deep sequencing study of the SLE case-control cohort, in a target gene approach. They incorporated ~2,000 SLE cases and controls. The target gene was DAP1. Authors identified suggestive associations of the DAP1 common variants on SLE risk. Expression quantitative trait (eQTL) effects.

1. This is not clear for the reviewer why the authors only focused on the DAP1 genes. Since a variety of SLE genes have been reported, there needs rationale to select DAP1, which this reviewer could not find.
Response: We have also sequenced other SLE risk loci in our previous study (1). In the present study, DAP1 was targeted for deep sequencing because this gene was implicated as a candidate gene in an earlier familial linkage study of SLE and RA (2, 3). Also, meta-analyses of GWAS datasets had associated genetic variations in DAP1 with ulcerative colitis and IBD (4, 5). Please see our comments to reviewer #1 also addressing this question. As both reviewers' suggested, we have now incorporated this information in the revised manuscript.

2. The merit of deep sequencing is not clear. The identified variants were common, and the typical GWAS and subsequent imputation can capture them, so no need to conduct deep sequencing.
Response: A 102 Kb genomic segment containing the DAP1 locus is represented by just 12 SNPs on Immunochipv.1, an illumina SNP Chip array that was extensively used in many Genomewide association studies (GWASs) on various immune system related diseases, including SLE. Deep sequencing of this interval with an average depth of 153.2-fold in present study identified 2094 high quality sequence variants, of which 1764 were low frequency (MAF <0.05) and 330 were common polymorphisms (Supplementary Table 2a and 2b). These results illustrate that the DAP1 genomic region is extremely polymorphic, having an average of 20.5 variants/kb of which 3.23 were common within the Caucasian population.

Sequencing captured 330 DAP1 variants with minor allele frequency of $\geq 5\%$ in study population, 39 of those showed statistically significant ($p < 0.05$, FDR) association with SLE in Caucasians. The sequencing identified DAP1 variants of functional significance. This existed with a 5-12% minor allele frequency in the population and importantly, was never captured using the previous SNP based arrays. For example, sequencing revealed SNP rs267985 [OR (95% CI) = 1.38 (1.1-1.6), $p = 1.81e-04$], the peak variants in Caucasians, which is an intronic variant with strong regulatory potential based on the ENCODE and RegulomeDB databases. In summary, the sequencing approach permits a fine mapping of the DAP1 locus and revealed low frequency functional variants that show association with SLE in multiple ethnic groups. Noteworthy, about 90% of the SLE risk haplotype variants that exhibit strong eQTLs were only discovered using our sequencing approach. In summary, the discovery of the disease associated, low frequency variants, was only possible in sequencing and not with SNP array. This really illustrates the importance of our sequencing. Notably, the sequencing data identified several DAP1 variants of functional significance. These will be explored in functional assays in future investigations on SLE and other diseases.

3. SLE risk variants are likely to have eQTL effects of white blood cells. So the findings were not novel.
Response: Many of the SLE associated eQTLs of DAP1 gene were not reported earlier, and we argue these data are novel. The functionally important DAP1 haplotype described in the present study was not reported previously. We confirmed the eQTL effect of this haplotype in 1KG data sets. Not all disease associated variant show eQTLs in white blood cells. But one needs to first capture all potential variants that could be causal or indirectly associated with disease. We argue that our approach is novel and robust, identifying variants that directly and indirectly link phenotype to disease. In addition, in our present study, more than 600 potentially functional rare DAP1 variants were discovered (Supplementary Table 2). Future functional studies will explore the impact of these variants on gene expression and susceptibility to SLE and other autoimmune and rheumatic diseases. Thus, in typical GWAS analysis, the array of variants detected by our sequencing approach were never detected. eQTL effects associated with DAP1 haplotypes were not reported earlier, nor assessed for association with disease. Our targeted sequencing approach captures all common and low frequency variants within a candidate LD block, thus allowing a precise assessment of genetic risk at this locus. We show that individual homozygotes for protective allele HAP1-HAP1 have significantly higher expression of DAP1 as compared to risk allele, HAP3-HAP2 and HAP3-HAP3 (Figure 2.G-H), a strong eQTL effect. So, we strongly believe that these findings are novel.

4. Significance of the identified association signals were not enough. Considering potential multiple comparisons on other genes, at least, the typical genomewide significance threshold is required ($P < 5 \times 10^{-8}$).

Response: Given the fact that SLE associated functional haplotype is a low frequency allele, it is obvious that larger sample size is required to reach typical genomewide level of significance. But nonetheless, our analytical strategy identified 39 common variants with a statistically significant association with SLE, with SNP rs267985 (OR (95% CI) = 1.38 (1.1-1.6), $p=1.81e-04$] having peak association in the EA discovery cohort. As we discussed our DAP1 manuscript (Supplemental Figure 5), Chochran-Mantel-Haenszel analysis of the total cohort of three independent ethnic groups detected the peak association with rs62337599 ($p < 8.26 \times 10^{-6}$) and identified three other variants reaching suggestive genomewide level of significance. We subsequently performed haplotype analysis on the 19 strongly associated functional variants within the DAP1 regulatory region and identified a common SLE risk haplotype (HAP3) in all three ethnic groups (OR=1.5, $p < 4.51e-05$) and a common SLE protective haplotype (HAP1) in all three ethnic groups (OR = 0.7, $p < 1.88e-06$). Thus, these data place DAP1 in the suggestive genomewide level of significance like ~90 other SLE risk loci that are considered as important SLE risk alleles.

Reviewer #3:

Following up on nominal association of rare genetic variants in an SLE Immunochip study, the author perform deep sequencing of the DNA surrounding the DAP1 gene in patients with SLE and controls of various ancestries. After identifying and replicating association of a rare haplotype not covered by the Immunochip, the author identify a robust eQTL (at mRNA and protein levels) and allelic autoantibody association. The genetic and biological conclusions of this study were replicated in independent cohorts of patients, which provides confidence in the significance and impact of this study. Despite reviewer enthusiasm for the findings of this manuscript, the format and lack of clarity in the supplemental materials was problematic. Altogether, this is an exciting manuscript that will benefit from extensive reformatting of the Tables and Supplements.

Major comments:

The authors identify a large variance in DAP1 expression levels that is in part explained by the SLE-risk haplotype. Is there a genotype-dependent change in the variance in DAP1 expression (as assessed using variance association mapping)? This analysis would also be particularly interesting in the autoantibody associations presented in Figure 3.

Response: Yes. DAP1 expression does show genotype dependent variation. As shown in Fig.2g & h, DAP1 gene expression is significantly high in protective genotype (Hap1-Hap1) as compared to SLE risk genotype (Hap3-Hap3) in LCLs and MDMs. We also observed association of antibody titers with DAP1 genotypes. See following plots. Figure 5 does show data on risk genotype and antibody association. Fig.5A show distribution of anti-nuclear antibodies (ANA) in three genotypes of one of the potential risk SNP rs2930047, and Fig.5E shows enrichment of anti-sm/SmD antibodies in individuals with DAP1 risk haplotype. In addition to that we generated additional data (Supplementary Fig. 7) as per reviewer's suggestion. As shown below in right panel, titers of several non-nuclear antigens were found higher in patients with risk haplotype than those with protective haplotypes. This is very interesting finding, given the fact that a fraction of SLE population does exhibit higher titers of antibodies targeting non-nuclear antigens.

In the introduction, the authors mention that a variant in DAP1 is associated with IBD at genome-wide significance, what is the exact relationship of the IBD-associated variant to the SLE association in terms of linkage disequilibrium?

Response: With respect to the reported association of DAP1 with IBD, the tagging SNP rs2930047 was associated with IBD in an extremely large meta-analysis study (80,064 cases and controls) at genome-wide significance levels with a combined p-Value of $1E-8$ (see below). IBD implicated SNP rs2930047 is a member of the 19 SNPs included in the SLE risk haplotype (SNP4) we defined for DAP1 in present study (Figure 1). IBD risk allele "C" of rs2930047 is the risk for SLE as well. As we have presented in Figure 2, rs2930047 is one of several SNPs in the risk haplotype that are annotated as strongly functional by ENCODE (Fig. 2C), and the risk allele of rs2930047 (for both SLE in our study and IBD) is

associated with downregulation of DAP1 transcription in public eQTLs (Fig. 2e and Fig. 2g), as well as in our own macrophage eQTL panel. The meta-analysis that identified an association of rs2930047 with IBD did not perform haplotype analysis, although LD is quite strong within the DAP1 LD block (shown in Fig. 1b) and it is quite likely that haplotype analysis of the IBD association would identify the same or a very similar risk haplotype to what we defined for SLE.

The authors report that 27% of variants in DAP1 mapped to known ENCODE transcription factor binding sites. Are any predicted to break or change the preferred TF binding site?

*Response: Overall, about 41% (868/2094) of DAP1 variants were annotated potentially functional, means they could impact binding of transcription factor and or have impact of gene expression (Supplementary Table 2). Based on RegulomeDB functional annotation (<https://regulomedb.org/regulome-search/>), about 5% (96/2094) were with predictions for eQTLs and strong likelihood of impacting the binding of important transcription factors or other molecules at this locus. The variants are highlighted with red color and bolded in Supplementary Table 2. RegulomeDB is a database that annotates SNPs with known and predicted regulatory elements in the intergenic regions of the *H. sapiens* genome. Known and predicted regulatory DNA elements include regions of DNase hypersensitivity, binding sites of transcription factors, and promoter regions that have been biochemically characterized to regulate transcription. A snapshot of Suppl. Table 2.*

What multiple testing correction was applied in the RNA-seq analysis presented in Sup Table 9? The term "average expression protection (6)" is not clear.

Response: Gene expression was corrected for multiple genes using MetLab tools. "Average expression protection" means average gene expression (RPKM values) in protective and SLE risk genotype. Methods for data normalization and multiple corrections are based on the use of "internal standards" that characterize some aspects of the system's behavior, such as technical variability, as presented elsewhere (Dozmorov and Centola 2003; Dozmorov and Lefkovits 2009; Dozmorov, Jarvis et al. 2011). Created initially for the analysis of microarray data they were slightly modified to the needs of RNA-seq data analysis.

Differential gene expression analysis (Dozmorov and Centola 2003; Dozmorov and Lefkovits 2009; Dozmorov, Jarvis et al. 2011) - includes the following steps:

Construction of the 'reference group' by identifying a group of genes expressed above background with inherently low variability as determined by an F-test. The 'reference group' presents an internal standard of equal expression. As such, the 'reference group' is used to assess the inherent variability resulting from technical factors alone (technological variation). By creating an estimate of the technological variation we are able to select a group of biologically stable genes.

Selection of replicates using the commonly accepted significance threshold of $p < 0.05$ with a Student T-test. This selection maintains the commonly accepted sensitivity level; however, a significant proportion of genes identified as differentially expressed at this threshold will represent false positive determinations.

An Associative T-test in which the replicated residuals for each gene from the experimental group are compared with the entire set of residuals from the reference group defined above. The H_0 hypothesis is checked to determine whether the levels of gene expression in the experimental group presented as replicated residuals (deviations from the averaged control group profile) is associated with a highly representative (several hundred members) normally distributed set of residuals of gene expression values in the reference group. The significance threshold is then corrected to render the appearance of false positive determinations improbable: in current case $p < 0.0001$. Only genes that pass both tests are presented in the final selections. Additional restrictions were applied to the minimal gene expression level (RPKM > 2) and fold of changes (> 1.5). The two-step normalization procedure and the Associative analysis functions are implemented in MatLab (Mathworks, MA). These methods have been published, see following articles.

*Dozmorov, I. and M. Centola (2003). "An associative analysis of gene expression array data." *Bioinformatics* 19(2): 204-211.*

*Dozmorov, I. and I. Lefkovits (2009). "Internal standard-based analysis of microarray data. Part 1: analysis of differential gene expressions." *Nucleic Acids Res* 37(19): 6323-6339.*

*Dozmorov, I. M., J. Jarvis, et al. (2011). "Internal standard-based analysis of microarray data--analysis of functional associations between HVE-genes." *Nucleic acids research* 39(18): 7881-7899.*

Minor comments (these are minor because they can be very easily addressed, but the formatting issues are extremely important for manuscript clarity and usefulness).

Sequencing of the DAP1 gene identified numerous rare and common variants, and the authors indicate that DAP1 is an extremely polymorphic gene. Is this region of the genome more polymorphic (statistically) than other regions encoding protein-coding genes (when accounting for gene length)?

Response: HLA gene is a previously known highly polymorphic locus in human genome. We assessed the number of variants per KB of the genome across several SLE associated loci and found that DAP1 is next to HLA in terms of number of variants per KB, suggesting that it is a polymorphic gene. Like HLA has been shown to be evolved in response to human diseases, especially infectious diseases. DAP1 polymorphisms may also be derived by certain selection pressure.

European/European American is preferable to Caucasian as the African American equivalent to Caucasian is socially insensitive.

Response: We will make this correction as suggested by reviewer.

The authors should use rsIDs or genomic position (chr:position) rather than new identifiers (such as SNP1, SNP2, SNP3) to allow subsequent studies to easily identify and assess these variants in their datasets.

Response: We apologize for the confusion. The phylogenetic network program allows a defined number of letters in the name string for resolution purpose. So that's why we used SNP number instead of Chr:position or rsIDs which are longer to accommodate in the label in network program. But we agree with reviewer, so to make it clear and easy for subsequent studies to identify and follow these variants, we have provided rsIDs column corresponding to SNP1, SNP2,... in Table 1 and Supplementary Table 3. See first two columns in snapshot below:

As presented in the Reviewer pdf, Table 1 was extremely hard to read and interpret (it spans several pages). RegulomedB annotations need to be defined in a Table legend. It was difficult to identify the reason some rows were highlighted in yellow.

Response: We apologize for the formatting issues. Originally, these datasets were submitted as excel files, it may have happened during generation of combined pdf on all the files. We will send these data in an excel format should the editor agree. Yellow highlights were to identify some of the key SNPs. We have provided RegulomeDB classification scheme in the table footnote as suggested by reviewer.

In Table 2, the table also spans multiple pages and is hard to review. The column rows need to indicate that a haplotype frequency is given. The Chi-Squared p-value should be limited to three significant digits, and the OR column should be formatted as to not give an OR for non-significant associations (e.g. the OR of 5.2 is not significant given a p-value of 0.08; similarly, if both the cases and controls lack a haplotype, the OR cannot equal 100).

Response: We again apologize for formatting issue. Originally, these datasets were submitted as excel files, it may have happened during generating combined pdf on all the files. We will make sure that reviewers receive these data in excel formats. We have provided OR for all significant and non-significant haplotypes in order to indicate the directionality of association. Although, we agree with reviewer that these are not statistically significant. Some of these haplotypes are very low in frequency or rare, so we have to show three digits after decimal in frequency columns. We have done these corrections in the revised table.

It would be helpful to add a supplemental figure showing the gene DAP1 in the context of the genetic variants associated with SLE.

Response: A Manhattan plot showing genetic association of DAP1 with SLE has been added to the Supplementary Figure 1. We hope this what reviewer suggested.

Supplemental Table 2a is unhelpful as presented - the reviewer was unable to assess this table in the current format. This should be a supplemental dataset that can be provided as a tab-delimited txt file or excel file.

We apologize for format issue. Originally, these datasets were submitted as excel files, it may have happened during generating combined pdf on all the files. We will make sure that reviewers receive these data in excel formats.

Supplemental Table 5 was hard to interpret as it also spanned two pages. Additionally, it was confusing way the ORs of specific Diplotypes were highlighted as they had non-significant p-values (p-values>0.05)

Response: We apologize for format issue. Originally, these datasets were submitted as excel files, it may have happened during generating combined pdf on all the files. We will make sure that reviewers receive these data in excel formats. We agree the associations were non-significant but it is due to small sample size, however the point was to show the directionality of the association even in case of comparisons that didn't reach statistical significance level.

Supplemental Table 6 should also be reformatted as a txt file.

Response: We apologize for format issue. We will provide text or excel file for this Table.

The motifs on page 481 are not formatted in an interpretable fashion.

Response: We apologize, this is also due to file format issue. We will request editor to provide excel file for these data.

Supplemental Table 7 needs to be reformatted - either with smaller txt or as a txt file.

Response: We apologize for format issue. We will provide text or excel file for this Table in revised manuscript.

Supplemental Table 10 needs to be reformatted - the table can be smaller to fit on one page and the legend needs to be formatted to be legible.

Response: We apologize for format issue. We will provide text or excel file for Suppl. Table 10 in revised manuscript.

In the reviewer pdf, Supplemental Table 11 does not have a legible title.

Response: We apologize for format issue. We will provide text or excel file for this Table in revised manuscript.

Supplemental Table 11 should be a txt file. The ratio and p-value columns should show three significant digits. A README tab should clearly identify the source of values provided.

As suggested by reviewer, necessary corrections has been made in the revised manuscript.

Again, we thank reviewers for taking time to review our work and for providing constructive inputs to improve the manuscript. Hope that reviewers will be satisfied with our responses. Thanks.

References

1. Raj P, Rai E, Song R, Khan S, Wakeland BE, Viswanathan K, et al. Regulatory polymorphisms modulate the expression of HLA class II molecules and promote autoimmunity. *Elife*. 2016;5.
2. Nath SK, Namjou B, Garriott CP, Frank S, Joslin PA, Kilpatrick J, et al. Linkage analysis of SLE susceptibility: confirmation of SLER1 at 5p15.3. *Genes Immun*. 2004;5(3):209-14.
3. Namjou B, Nath SK, Kilpatrick J, Kelly JA, Reid J, James JA, et al. Stratification of pedigrees multiplex for systemic lupus erythematosus and for self-reported rheumatoid arthritis detects a systemic lupus erythematosus susceptibility gene (SLER1) at 5p15.3. *Arthritis Rheum*. 2002;46(11):2937-45.
4. Di Narzo AF, Peters LA, Argmann C, Stojmirovic A, Perrigoue J, Li K, et al. Blood and Intestine eQTLs from an Anti-TNF-Resistant Crohn's Disease Cohort Inform IBD Genetic Association Loci. *Clin Transl Gastroenterol*. 2016;7(6):e177.
5. Anderson CA, Boucher G, Lees CW, Franke A, D'Amato M, Taylor KD, et al. Meta-analysis identifies 29 additional ulcerative colitis risk loci, increasing the number of confirmed associations to 47. *Nat Genet*. 2011;43(3):246-52.
6. Koren I, Reem E, Kimchi A. Autophagy gets a brake: DAP1, a novel mTOR substrate, is activated to suppress the autophagic process. *Autophagy*. 2010;6(8):1179-80.
7. Koren I, Reem E, Kimchi A. DAP1, a novel substrate of mTOR, negatively regulates autophagy. *Curr Biol*. 2010;20(12):1093-8.

Second round of review

Reviewer 1

"Raj et al performed deep sequencing of the DAP1 genetic locus to evaluate association with SLE risk, in a modest multi-ethnic cohort of SLE patients and controls. A low frequency risk haplotype was identified, which is correlated with decreased expression of DAP1 mRNA and protein and with other clinical characteristics like autoantibody production. More details are needed to justify and understand the study design. It seems that DAP1 was resequenced as part of a previous, larger study that aimed at fine mapping candidate SLE associated loci, but it is not clear if results presented in the current manuscript were already described in the previous study. Previous genetic evidence of association with SLE was weak for this locus, what was the rationale to prioritise this region over others? Also, there is no detail on how the DAP1 region was defined and what are coordinates of this resequenced region."

The extended details of the region that was selected for resequencing should have been included in the revised manuscript.

"Figure 1b shows LD patterns for a region that is larger than the targeted re-sequenced DAP1 locus; where is the rest of the data shown coming from? What population was used to plot the LD patterns shown in Fig1C? rs267985 is the top associated SNP in Caucasians only."

Details provided should be included in the text/figure legend

"The authors claim that the risk SNP is correlated with decreased expression of DAP1 in B cells, but this is not supported by the results found on primary B cells shown on Supplementary Figure 3, this should be discussed."

The sentence "Our data on primary B cells of healthy donors also confirm low DAP1 expression in risk allele at baseline and after stimulation through various pathways (Supplementary Fig. 9)." Should still be rephrased

"Why was the transcriptome analysis performed in a heterogeneous cell population such as PBMCs? What was the individual cell types distribution in patients with the risk alleles compared to the patients with the protective alleles? Are the p value reported for differentially expressed genes corrected for multiple testing? Autophagy was not one of the top differential pathways in this analysis (figure 4); did the authors expect to see a difference?"

These limitations/issues have not been included in the discussion in the revised manuscript.

Reviewer 3

The authors have addressed my queries and concerns adequately.